# Iterative Distillation for Reward-Guided Fine-Tuning of Diffusion Models in Biomolecular Design

**Xingyu Su**[1][*]  **Xiner Li**[1][*]  **Masatoshi Uehara**[2][*][†]  **Sunwoo Kim**[3]  **Yulai Zhao**[4]
**Gabriele Scalia**[5]  **Ehsan Hajiramezanali**[5]  **Tommaso Biancalani**[5]
**Degui Zhi**[6]  **Shuiwang Ji**[1][‡]
[1]Texas A&M University  [2]EvolutionaryScale  [3]Seoul National University
[4]Princeton University  [5]Genentech  [6]University of Texas Health Science Center at Houston
`{xingyu.su,sji}@tamu.edu`

## Abstract

We address the problem of fine-tuning diffusion models for reward-guided generation in biomolecular design. While diffusion models have proven highly effective in modeling complex, high-dimensional data distributions, real-world applications often demand more than high-fidelity generation, requiring optimization with respect to potentially non-differentiable reward functions such as physics-based simulation or rewards based on scientific knowledge. Although RL methods have been explored to fine-tune diffusion models for such objectives, they often suffer from instability, low sample efficiency, and mode collapse due to their on-policy nature. In this work, we propose an iterative distillation-based fine-tuning framework that enables diffusion models to optimize for arbitrary reward functions. Our method casts the problem as policy distillation: it collects off-policy data during the roll-in phase, simulates reward-based soft-optimal policies during roll-out, and updates the model by minimizing the KL divergence between the simulated soft-optimal policy and the current model policy. Our off-policy formulation, combined with KL divergence minimization, enhances training stability and sample efficiency compared to existing RL-based methods. Empirical results demonstrate the effectiveness and superior reward optimization of our approach across diverse tasks in protein, small molecule, and regulatory DNA design. The source code is released at (`https://divelab.github.io/VIDD/`).

## 1 Introduction

Diffusion models (Sohl-Dickstein et al., 2015; Ho et al., 2020; Song et al., 2020) have achieved remarkable success across diverse domains, including computer vision and scientific applications (*e.g.*, protein design (Watson et al., 2023; Alamdari et al., 2023)). Their strength lies in modeling complex, high-dimensional data distributions, including natural images and chemical structures such as proteins and small molecules. However, in many real-world scenarios, especially for biomolecular design, generating samples that merely resemble the training distribution is not sufficient. Instead, we often seek to optimize specific downstream reward functions. For instance, in protein design, beyond generating plausible structures, practical applications frequently require satisfying task-specific objectives such as structural constraints, binding affinity, and hydrophobicity (Hie et al., 2022; Pacesa et al., 2024). To meet these requirements, fine-tuning diffusion models with respect to task-specific rewards is crucial, enabling goal-directed generation aligned with downstream objectives.

Numerous algorithms have been proposed for fine-tuning diffusion models with respect to reward functions, motivated by the observation that this problem can be naturally framed as a reinforcement learning (RL) task within an entropy-regularized Markov Decision Process (MDP), where each

---

[*]Equal contribution

[†]This work was done when he was at Genentech

[‡]Corresponding author

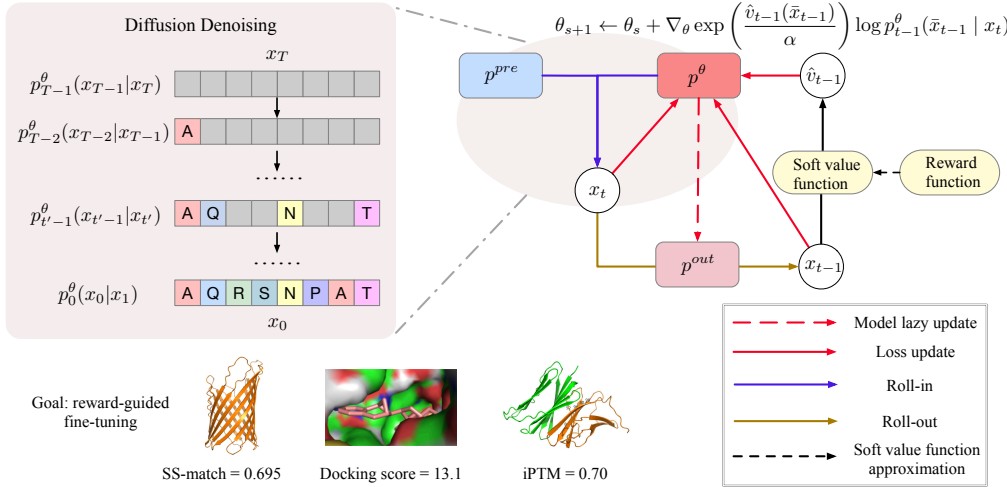

Figure 1: Overview of **VIDD**. **VIDD** fine-tunes diffusion models to maximize potentially non-differentiable rewards by iteratively distilling soft-optimal denoising policies. It alternates between (1) off-policy roll-in, (2) value-guided reward-weighted roll-out, and (3) forward KL-based model updates. Our algorithm leverages off-policy roll-ins and forward KL minimization rather, which contribute to improved optimization stability.

policy corresponds to the denoising process of the diffusion model (Black et al., 2024; Fan et al., 2023). In computer vision, current state-of-the-art methods fine-tune diffusion models by directly backpropagating reward gradients through the generative process (Clark et al., 2023; Prabhudesai et al., 2023). However, in many scientific applications, reward functions are inherently non-differentiable, making such optimization inapplicable. For example, in protein design, rewards based on secondary structure matching (e.g., DSSP algorithm (Kabsch & Sander, 1983)) or binding affinity predictions (e.g., AlphaFold3 (Abramson et al., 2024)) typically rely on hard lookup-tables based on scientific knowledge. Similarly, in small molecule design, reward functions such as synthetic accessibility (SA), molecular fingerprints (Yang et al., 2021), and outputs from physics-based simulators (e.g., AutoDock Vina (Trott & Olson, 2010)) are also non-differentiable. These characteristics pose a fundamental challenge for direct back-propagation approaches in scientific domains.

In such cases, policy gradient methods like Proximal Policy Optimization (PPO) (Schulman et al., 2017) offer a natural alternative, as they are used in diffusion models. However, PPO is known to exhibit instability, hyperparameter sensitivity, and susceptibility to mode collapse in many contexts Yuan et al. (2022); Moalla et al. (2024). These issues arise in part due to its on-policy nature — trajectories used for training the model are generated by the current fine-tuned policy, leading to narrow exploration around previously visited regions. Furthermore, from a theoretical perspective, PPO can be viewed as minimizing the *reverse* Kullback–Leibler (KL) divergence between the target and generated trajectory distributions. This reverse KL objective encourages mode-seeking behavior, potentially leading to mode collapse (Wang et al., 2023; Go et al., 2023; Kim et al., 2025).

To address the aforementioned challenges, we propose a new framework, **VIDD** (**V**alue-guided **I**terative **D**istillation for **D**iffusion models), designed to maximize possibly non-differentiable rewards in a stable and effective manner. The core idea is to iteratively distill soft-optimal policies—including, which serve as target denoising processes that optimize the reward while remaining close to the current fine-tuned model, as visualized in Figure 1. Concretely, the algorithm proceeds in three iterative steps: (1) roll-in using sufficiently exploratory off-policy trajectories, (2) roll-out to simulate soft-optimal policies, and (3) update the fine-tuned model by minimizing the KL divergence between the soft-optimal and current model policies. Importantly, in (2) and (3), our algorithm effectively leverages value functions tailored to diffusion models to guide fine-tuning, analogous to value-weighted MLE in RL (Peters et al., 2010). Notably, our framework leverages off-policy roll-ins—decoupling data collection from policy updates—and employs forward KL minimization rather than reverse KL, both of which contribute to improved optimization stability over complex reward landscapes.

Our contributions are summarized as follows. We propose a novel algorithm, **VIDD**, for fine-tuning diffusion models through iterative distillation of target policies composed of both value functions and the current policy. Unlike direct reward backpropagation methods, which require differentiable reward signals, our approach can handle non-differentiable rewards commonly encountered in scientific domains. Algorithmically, our key innovation lies in effectively incorporating value functions specifically tailored to the diffusion models. Empirically, we validate the effectiveness of our method across a range of scientific tasks, particularly in protein and small molecular design.

## 1.1 RELATED WORKS

**Fine-tuning diffusion models for reward-maximization.** When reward functions are differentiable, as is often the case in computer vision, state-of-the-art methods achieve strong performance by directly backpropagating gradients induced by the reward functions on the diffusion model (Clark et al., 2023; Prabhudesai et al., 2023; Wu et al., 2024; Wang et al., 2024). However, in many scientific domains such as biology and molecule field, reward functions are often non-differentiable (Lisanza et al., 2024; Hie et al., 2022). In such settings, a natural approach is to adopt RL techniques that optimize reward without requiring differentiability (Zhang et al., 2024; Gupta et al., 2025; Ektefaie et al., 2024; Vogt et al., 2024; Deng et al., 2024; Venkatraman et al., 2024; Rector-Brooks et al., 2024; Zekri & Boullé, 2025). For instance, DPOK (Fan et al., 2023) and DDPO (Black et al., 2024) adapt PPO, a stabilized variant of policy gradient methods, to the diffusion model fine-tuning. However, these methods often suffer from instability when applied to diffusion models (Clark et al., 2023, Figure 3), due to their inherently on-policy nature and the use of a reverse KL divergence objective between the fine-tuned policy and a target policy (discussed further in Section 5). In contrast, our method avoided on-policy updates and instead leverages a forward KL divergence objective, which stabilizes training and mitigates mode collapse.

**Inference-time technique for reward-maximization.** An alternative line of work focuses on training-free techniques that aim to improve rewards solely at inference time. The most straightforward example is Best-of-N sampling, which selects the highest-reward output from N generated candidates. More sophisticated methods go further by evaluating and selecting promising intermediate generation states during the sampling process (Wu et al., 2023; Li et al., 2024; Kim et al., 2025; Singhal et al., 2025; Ma et al., 2025; Tang et al., 2025). While these methods avoid the need for fine-tuning, they often incur significantly higher inference-time costs due to repeated sampling. Although they can boost reward in some cases, they do not directly improve the underlying generative model. In this respect, training-free methods are orthogonal and complementary to fine-tuning-based approaches, and the two can be effectively combined to achieve even stronger performance.

## 2 PRELIMINARY

We begin by reviewing the foundations of diffusion models. We then formulate our objective: fine-tuning diffusion models to maximize task-specific reward functions.

## 2.1 DIFFUSION MODELS

Diffusion models (Sohl-Dickstein et al., 2015; Ho et al., 2020; Song et al., 2020) aim to learn a sampling distribution $p(\cdot) \in \Delta(\mathcal{X})$ over a predefined design space $\mathcal{X}$ based on the observed data. Formally, a diffusion model defines a forward *noising* process $q_t(x_t \mid x_{t-1})$, which gradually corrupts from clean data $x_0 \sim p(x_0)$ over discrete time steps $t = 1, \ldots, T$ into noise. The learning objective is to approximate the reverse *denoising* process $p_{t-1}(x_{t-1} \mid x_t)$, where each $p_t$ is a kernel mapping from $\mathcal{X}$ to $\Delta(\mathcal{X})$, such that the overall reverse trajectory transforms noise samples back into data samples drawn from the true distribution. In practice, each $p_t$ is parameterized by a neural network and trained to minimize a variational lower bound on the negative log-likelihood of the data.

**Notation.** With slight abuse of notation, we often denote the initial noise distribution $p_T \in \Delta(\mathcal{X})$ as $p_T(\cdot|\cdot)$, and we often refer to the denoising process as a *policy*, following terminology commonly used in RL. We denote by $\hat{x}_0(x_t) : \mathcal{X} \to \mathcal{X}$ the neural network used in pretrained diffusion models to predict the denoised input. Extension to other parameterizations is straightforward.

## 2.2 REWARD MAXIMIZATION IN DIFFUSION MODELS

Our goal is to fine-tune diffusion models to produce outputs that achieve high rewards. Here, we formalize the problem and highlight key challenges.

**Task Description**  Our goal is to fine-tune a pretrained diffusion model to generate samples that maximize a task-specific reward function. Formally, given a pretrained diffusion model $p^{\mathrm{pre}} \in \Delta(\mathcal{X})$ and a reward function $r : \mathcal{X} \to \mathbb{R}$, we aim to fine-tune pre-trained models such that

$$\underset{\theta}{\mathrm{argmax}}\ \underbrace{\mathbb{E}_{x_0 \sim p^\theta}[r(x_0)]}_{(a)\textbf{reward maximization}}\ -\alpha \underbrace{\mathrm{Dis}(\{p^\theta\}, \{p^{\mathrm{pre}}\})}_{(b)\textbf{regularization}},$$

where $p^\theta \in \Delta(\mathcal{X})$ denotes the distribution induced by the fine-tuned reverse denoising process $\{p_t^\theta\}_{t=T}^0$, and $p^{\mathrm{pre}}$ denotes the original pretrained distribution. Term (a) promotes the generation of high-reward samples, while term (b) penalizes deviation from the pretrained model to maintain the naturalness of generated samples. For example, when the discrepancy measure **Dis** in (b) is the KL divergence, the optimal solution takes the following form up to normalizing constant:

$$\exp(r(\cdot)/\alpha)p^{\mathrm{pre}}(\cdot) \tag{1}$$

**Challenges.**  In domains such as computer vision, reward functions are typically modeled using differentiable regressors or classifiers, allowing for direct gradient-based optimization (Clark et al., 2023; Prabhudesai et al., 2023). In contrast, many key objectives in biomolecular design, such as structural constraints, hydrophobicity, binding affinity, often rely on non-differentiable features, as detailed in Section 1. While RL-based approaches, such as policy gradient methods, can in principle handle non-differentiable feedback (see Section 1.1), they often suffer from instability due to their on-policy nature and reliance on reverse KL divergence. In the following, we introduce a novel fine-tuning method that can effectively optimize possibly non-differentiable rewards.

## 3 ITERATIVE DISTILLATION FRAMEWORK FOR FINE-TUNING DIFFUSION MODELS

Our algorithm is motivated by the goal of distilling desirable teacher policies that maximize task-specific reward functions. To clarify this motivation, we first define what constitutes teacher policies in our setting, followed by a detailed description of our approach to distilling them.

### 3.1 TEACHER POLICIES IN DISTILLATION

We introduce *soft-optimal policies*, which serve as the teacher policies that our algorithm aims to distill. Specifically, we define the soft-optimal policy $p_{t-1}^\star : \mathcal{X} \to \Delta(\mathcal{X})$ as a value-weighted variant of the pre-trained policy:

$$p_{t-1}^\star(\cdot|x_t) = \frac{p_{t-1}^{\mathrm{pre}}(\cdot|x_t)\exp(v_{t-1}(\cdot)/\alpha)}{\exp(v_t(x_t)/\alpha)}. \tag{2}$$

Here, for $t \in [T+1, \cdots, 1]$, the soft value function is defined as:

$$v_{t-1}(\cdot) := \alpha \log \mathbb{E}_{x_0 \sim p^{\mathrm{pre}}(x_0|x_{t-1})}\left[\exp\left(\frac{r(x_0)}{\alpha}\right)|x_{t-1} = \cdot\right], \tag{3}$$

where the expectation is taken under the trajectory distribution induced by the pre-trained policies.

These soft-optimal policies naturally arise when diffusion models are framed within entropy-regularized Markov Decision Processes. A key property of these policies is that, when sampling trajectories according to the soft-optimal policy sequence $\{p_t^\star\}_{t=T}^0$, the resulting marginal distribution over final outputs approximates the target distribution $\exp(r(\cdot)/\alpha)p^{\mathrm{pre}}(\cdot)$ in (1) (Uehara et al., 2025a).

Importantly, many test-time guidance methods—such as classifier guidance in both continuous diffusion models (e.g., (Dhariwal & Nichol, 2021; Bansal et al., 2023)) and discrete diffusion models

(e.g., (Nisonoff et al., 2024)), as well as sequential Monte Carlo (SMC)-based approaches (e.g., (Wu et al., 2023; Li et al., 2024; Kim et al., 2025))—can be interpreted as approximate sampling schemes for these soft-optimal policies (Uehara et al., 2025a). While test-time guidance methods are appealing due to their ease of implementation, they often incur substantial computational overhead during inference and may struggle to achieve consistently high reward values. In contrast, we explore how soft-optimal policies can inform the fine-tuning of diffusion models, enabling reward-guided generation without requiring any additional computation at test time.

## 3.2 ITERATIVE DISTILLATION

Thus far, we have introduced soft-optimal policies and discussed their desirable properties. Within our framework, we designate these policies as *teacher policies*, while the fine-tuned models are treated as *student policies* (Czarnecki et al., 2019). A natural objective for distilling such policies into a fine-tuned model $\{p_t^\theta\}$ is given by

$$\underset{\theta}{\arg\min} \sum_t \mathbb{E}_{x_t \sim u_t}[\mathrm{KL}(p_{t-1}^\star(\cdot|x_t)\|p_{t-1}^\theta(\cdot|x_t))], \tag{4}$$

where $u_t \in \Delta(\mathcal{X})$ denotes a roll-in distribution, where we will elaborate on in Section 4.1. By simple algebra (see the detailed derivation in Appendix B), up to some constant, this objective reduces to

$$\underset{\theta}{\arg\max} \sum_t \mathbb{E}_{x_t \sim u_t} \left[ \frac{1}{\exp(v_t(x_t)/\alpha)} \mathbb{E}_{x_{t-1} \sim p_{t-1}^{\mathrm{pre}}(x_{t-1}|x_t)}[\exp(v_{t-1}(x_{t-1})/\alpha) \log p_{t-1}^\theta(x_{t-1}|x_t))] \right]. \tag{5}$$

As we will demonstrate in Section 4.3, this objective can be optimized in practice by approximating value functions and replacing expectations with sample estimates. Notably, the objective is off-policy in nature, meaning that the roll-in distribution $u_t$ can be arbitrary and does not need to match the current model policy. Indeed, this objective, commonly referred to as value-weighted maximum likelihood in the RL literature (Peters et al., 2010), has been employed as a scalable and stable approach for off-policy RL (Peng et al., 2019).

In practice, we adopt an iterative distillation procedure. This is motivated by the fact that, due to the reliance on empirical samples and approximate value functions, it is generally infeasible to accurately distill the target policy $p_{t-1}^\star$ into the student policy $p_{t-1}^\theta$ in a single step. Rather than fixing $p_{t-1}^{\mathrm{pre}}$ throughout training, we periodically update it by the latest student policy but in a lazy manner, allowing the target to gradually improve over time. This dynamic yet infrequent (lazy) target update enables progressive refinement, allowing both the teacher and student policies to evolve iteratively toward better alignment. This strategy is analogous to standard RL practices grounded in the Policy Improvement Theorem (Mnih et al., 2015).

To formalize the above iterative process, we introduce an iteration index $s \in [1, \cdots, S]$ and perform recursive updates as follows:

$$\theta_{s+1} \leftarrow \theta_s + \nabla \sum_t \mathbb{E}_{x_t \sim u_t, p_{t-1}^{\mathrm{out}}(x_{t-1}|x_t)} \left[ \frac{\exp(v_{t-1}^{\mathrm{out}}(x_{t-1})/\alpha)}{\exp(v_t(x_t)/\alpha)} \log p_{t-1}^\theta(x_{t-1}|x_t) \right]. \tag{6}$$

where $\{p^{\mathrm{out}}\}$ denotes the roll-out policy, and $v_{t-1}^{\mathrm{out}}$ is the corresponding soft value function. The roll-out policy is updated in a lazy manner—specifically, it is refreshed every $K$ steps using the current model parameters $p^{\theta_s}$. This lazy update scheme is critical for algorithmic stability in the off-policy setting, preventing rapid changes in the target while still allowing gradual improvement of the student policy toward higher reward. In the next section, we formalize this iterative distillation process as a complete algorithmic procedure.

## 4 ITERATIVE VALUE-WEIGHTED MLE

In this section, we present our algorithm for fine-tuning diffusion models to maximize reward functions. The full procedure is summarized in Algorithm 1. Each training iteration consists of three key components: (1) the *roll-in* phase, which defines the data distribution over which the loss is computed; (2) the *roll-out* phase, which aims to approximate the teacher policy by sampling from a

---

**Algorithm 1 VIDD** (**V**alue-guided **I**terative **D**istillation for **D**iffusion models)

---

1: **Require**: reward $r : \mathcal{X} \to \mathbb{R}$, pretrained model $\{p_{t-1}^{\text{pre}}(\cdot \mid x_t)\}$, fine-tuned model $\{p_{t-1}^{\theta}(\cdot \mid x_t)\}$

2: Initialize fine-tuned model $p^{\theta} = p^{\text{pre}}$, roll-out model $p^{\text{out}} = p^{\text{pre}}$, lazy update interval $K$, $\alpha$

3: **for** $s \in [1, \cdots, S]$ **do**

4:     **// Roll-in phase**

5:     Generate roll-in samples following roll-in policies (explain in Section 4.1) and collect $\mathcal{D} = \{x_T^{(i)}, \cdots, x_0^{(i)}\}_{i=1}^N$.

6:     **// Roll-out phase**

7:     Obtain a sample following the roll-out policy $p_{t-1}^{\text{out}}$: $\bar{x}_{t-1}^{[i]} \sim p_{t-1}^{\text{out}}(x_{t-1}^{[i]} \mid x_t^{[i]})$ for each $t$.

8:     Approximate soft-value functions $\hat{v}_{t-1}(\bar{x}_{t-1}^{[i]}) \leftarrow r\left(\hat{x}_0(\bar{x}_{t-1}^{[i]}; \theta^{\text{out}})\right)$ and $\hat{v}_t(x_t^{[i]}) \leftarrow r\left(\hat{x}_0(x_t^{[i]}; \theta^{\text{out}})\right)$.

9:     If $S \% K = 0$, update the roll-out policy: $\{p_t^{\text{out}}\} \leftarrow \{p_t^{\theta_s}\}$ and $\theta^{\text{out}} \leftarrow \theta_s$.

10:     **// Distillation phase**

11:     Update model parameters $\theta$ by gradient ascent:

$$\theta_{s+1} \leftarrow \theta_s + \gamma \nabla_\theta \sum_{i=1}^N \sum_t \left( \frac{\exp(\hat{v}_{t-1}(\bar{x}_{t-1}^{[i]})/\alpha)}{\exp(\hat{v}_t(x_t^{[i]})/\alpha)} \log p_{t-1}^{\theta}(\bar{x}_{t-1}^{[i]} \mid x_t^{(i)}) \right). \tag{7}$$

12: **end for**

13: **return** $\theta_S$

---

roll-out policy and computing its corresponding weight (soft value); and (3) the *distillation* phase, where the objective is defined as the KL divergence between the teacher policies and the student policies (i.e., the fine-tuned models). We detail each of these components below.

## 4.1 ROLL-IN PHASE

Due to the off-policy nature of our algorithm, we have significant flexibility in selecting the roll-in distribution path $x_T, x_{T-1}, \cdots, x_0$, which subsequently serves as the training distribution for loss computation. A well-designed roll-in policy should satisfy two competing desiderata: (1) exploration—ensuring broad coverage over the design space to avoid local optima, and (2) exploitation—maintaining proximity to high-reward trajectories for efficient policy improvement.

To balance these goals, we adopt a mixture roll-in strategy by sampling the roll-in distribution from:

- the pre-trained policy $p_t^{\text{pre}}$, which promotes exploration by generating diverse trajectories;
- the roll-out policy $p_t^{\text{out}}$ which is periodically updated and reflects stabilized knowledge from the student model.

At each training step, we sample from $p_t^{\text{pre}}$ with probability $1 - \beta_s$, and from $p_t^{\text{out}}$ with probability $\beta_s$. Further details on the construction of $p_t^{\text{out}}$ are provided in the following subsection.

## 4.2 ROLL-OUT PHASE

Our goal in this step is to approximate the current teacher policy. To this end, we aim to (1) sample $x_{t-1}$ conditioned on $x_t$ for each timestep $t$ following a roll-out policy, and (2) compute its corresponding soft-value, which will later serve as a weight during the distillation process.

**Sampling from roll-out policy.** Recall the motivational formulation in Equation (6). Our goal here is to replace the expectation with its empirical counterpart. To this end, we draw samples $\bar{x}_{t-1}$ from $p_{t-1}^{\text{out}}$, which is updated periodically at fixed time intervals.

**Approximation of soft value functions.** Recall that soft-value functions in (3) are defined as conditional expectation given $\bar{x}_{t-1}$. While one could estimate these expectations using Monte

Carlo sampling or regression, as commonly done in standard RL settings, we recommend a more practical approximation: $\hat{v}_{t-1}(\bar{x}_{t-1}) := r(\hat{x}_0(\bar{x}_{t-1}; \theta^{\text{out}}))$. Recalling $\hat{x}_0(\bar{x}_{t-1}; \theta^{\text{out}})$ is the denoised prediction from the diffusion model parameterized by the current student policy. This approximation is based on the replacement of the expectation in (3) with its posterior mean. We apply the same approximation to $\hat{v}_t(x_t)$, estimating it via $r(\hat{x}_0(x_t; \theta^{\text{out}}))$. Further discussions on soft value functions can be found in Appendix F.

Notably, this approximation has been implicitly adopted in several recent test-time reward optimization methods (Chung et al., 2022; Wu et al., 2023; Li et al., 2024). Building on its empirical success, we extend this idea to the fine-tuning setting. Compared to Monte Carlo-based methods, this approximation is computationally efficient—requiring only a single forward pass through the denoising network—and avoids the need to train an additional value function.

## 4.3 DISTILLATION PHASE

Thus far, we have defined the training data distribution via the roll-in phase and specified the supervised signal through the roll-out phase. The final step is to perform distillation by solving a supervised learning problem over the roll-in distribution. Specifically, we minimize the KL divergence between the teacher and student policies, as formalized in (6). In our implementation, soft value functions are approximated using estimates from trajectories. The expectations over roll-in and roll-out distributions are replaced with empirical estimates obtained from samples collected in their respective phases. The resulting loss function is given in (7), which corresponds precisely to a value-weighted maximum likelihood objective.

## 5 COMPARISON WITH POLICY GRADIENT METHODS

In this section, we highlight the key differences between our algorithm, **VIDD**, and policy gradient methods (Fan et al., 2023; Black et al., 2024). Broadly, there are two main distinctions: (1) policy gradient methods are inherently on-policy, whereas **VIDD** naturally supports off-policy updates; and (2) policy gradient methods implicitly optimize the reverse KL divergence between the fine-tuned policy and the soft-optimal policy, while our objective more closely aligns with the forward KL divergence. We elaborate on each of these differences below.

**On-policy vs. off-policy nature.** In policy gradient algorithms, fine-tuning is typically framed as the optimization of the following objective:

$$J(\theta) = \mathbb{E}_{\{p_t^\theta\}_{t=T}^0} \left[ r(x_0) - \alpha \sum_{t=T+1}^1 \text{KL}(p_{t-1}^\theta(\cdot \mid x_t) \| p_{t-1}^{\text{pre}}(\cdot \mid x_t)) \right].$$

In practice, this objective is optimized using the policy gradient (PG) theorem. For instance, when $\alpha = 0$, the gradient simplifies to the standard form (Black et al., 2024):

$$\nabla J(\theta) = \sum_t \mathbb{E}_{\{p_t^\theta\}_{t=T}^0} \left[ r(x_0) \nabla \log p_{t-1}^\theta(\cdot | x_t) \right]. \tag{8}$$

However, accurate gradient estimation requires that the roll-in distribution used to sample $x_t$ matches the current policy, making the method inherently on-policy. This constraint limits exploration and increases the risk of convergence to suboptimal local minima. Even more stable variants, such as PPO, remain sensitive to hyperparameter choices (Adkins et al., 2024), and overly strong regularization—while stabilizing the landscape—can significantly slow down learning (Clark et al., 2023, Fig. 3).

In contrast, **VIDD** naturally accommodates off-policy updates, allowing the use of more exploratory roll-in distributions without sacrificing training stability.

**Reverse KL vs. forward KL divergence minimization.** Furthermore, we show that the PPO objective $J(\theta)$ is equivalent to minimizing the reverse KL divergence between the trajectory distributions induced by the soft-optimal policy and those induced by the fine-tuned policy.

**Theorem 1.** *Denote $p_{0:T}^\theta \in \mathcal{X} \times \cdots \mathcal{X}$ as the induced joint distribution from $t = T$ to $t = 0$ by $\{p_t^\theta\}$ and denote $p_{0:T}^\star$ as the corresponding distribution induced by the soft-optimal policy. Then,*

$$J(\theta) = \text{KL}(p_{0:T}^\theta(\cdot) \| p_{0:T}^\star(\cdot)) = \sum_t \mathbb{E}_{\{p_t^\theta\}_{t=T}^0} [\text{KL}(p_{t-1}^\theta(\cdot | x_t) \| p_{t-1}^\star(\cdot | x_t))].$$

Table 1: Performance of different methods on both protein, DNA, and molecular generation tasks w.r.t. rewards and naturalness. The best result among fine-tuning baselines is highlighted in **bold**. We report the 50% quantile of the metric distribution. The $\pm$ specifies the the standard error of the estimate quantile with 95% confidence interval.

| Method | Protein SS-match | | | DNA Enhancer HepG2 | | | Molecule Docking - Parp1 | |
|---|---|---|---|---|---|---|---|---|
| | $\beta$-sheet%↑ | pLDDT↑ | Diversity↑ | Pred-Activity↑ | ATAC-Acc↑ | 3-mer Corr↑ | Docking Score↑ | NLL↓ |
| Pre-trained | $0.05 \pm 0.05$ | $0.37 \pm 0.09$ | **0.91** | $0.14 \pm 0.26$ | $0.000 \pm 0.000$ | -0.15 | $7.2 \pm 0.5$ | $971 \pm 32$ |
| Best-of-N (N=32) | $0.26 \pm 0.13$ | $0.38 \pm 0.11$ | 0.90 | $1.30 \pm 0.64$ | $0.000 \pm 0.000$ | -0.17 | $10.2 \pm 0.4$ | $951 \pm 22$ |
| DRAKES | - | - | - | $6.44 \pm 0.04$ | $\textbf{0.825} \pm \textbf{0.028}$ | 0.307 | - | - |
| Standard Fine-tuning | $0.48 \pm 0.16$ | $0.30 \pm 0.04$ | 0.57 | $1.17 \pm 1.23$ | $0.094 \pm 0.292$ | 0.829 | $7.8 \pm 1.8$ | $908 \pm 77$ |
| DDPP | $0.63 \pm 0.07$ | $0.36 \pm 0.07$ | 0.85 | $5.33 \pm 0.94$ | $0.305 \pm 0.460$ | **0.879** | $7.9 \pm 1.3$ | $981 \pm 52$ |
| DDPO | $0.81 \pm 0.02$ | $0.55 \pm 0.05$ | 0.52 | $7.38 \pm 0.11$ | $0.086 \pm 0.280$ | 0.398 | $8.5 \pm 1.3$ | $929 \pm 43$ |
| VIDD | $\textbf{0.83} \pm \textbf{0.01}$ | $\textbf{0.82} \pm \textbf{0.01}$ | 0.52 | $\textbf{8.28} \pm \textbf{0.18}$ | $0.820 \pm 0.384$ | 0.162 | $\textbf{9.4} \pm \textbf{1.7}$ | $\textbf{741} \pm \textbf{21}$ |

In contrast, **VIDD** optimizes an objective that more closely resembles the forward KL divergence. Since the reverse KL is known to be mode-seeking and can lead to unstable optimization landscapes (Wang et al., 2023; Go et al., 2023; Kim et al., 2025), avoiding reverse KL minimization contributes to more stable and effective fine-tuning.

## 6 EXPERIMENTS

Thus far, we have introduced **VIDD**, a framework designed to optimize possibly non-differentiable downstream reward functions effectively and stably in diffusion models in a sample-efficient manner. In this section, we evaluate the performance of **VIDD** across a range of biomolecular design tasks. We begin by describing the experimental setup.

### 6.1 EXPERIMENTAL SETUP

#### 6.1.1 TASK DESCRIPTORS

We aim to fine-tune diffusion models by maximizing task-specific reward functions. In the following, we outline the choice of the pre-trained diffusion models and the formulation of the reward functions used in our biomolecular design tasks (protein, DNA, small molecule design).

**Protein sequence design.** We adopt EvoDiff (Alamdari et al., 2023) as our pre-trained diffusion model, a representative masked discrete diffusion model for protein sequence design, trained on the UniRef database (Suzek et al., 2007). We use EvoDiff as an unconditional generative model to produce natural protein sequences. To tackle downstream protein design tasks, we use the following reward functions inspired by prior work (Hie et al., 2022; Verkuil et al., 2022; Lisanza et al., 2024; Uehara et al., 2025b; Pacesa et al., 2024). Appendix D.1 provides detailed definitions of each reward function. For tasks involving secondary structure matching optimization, we employ ESMFold (Lin et al., 2023) to predict the 3D structures of generated sequences. For protein binder design, we leverage AlphaFold2 (Jumper et al., 2021) to model the 3D structure of the multimers. In order to encourage the naturalness and foldability of the designed proteins, we additionally incorporate structural confidence metrics, such as pLDDT and radius of gyration (Pacesa et al., 2024), as part of the reward function.

- **ss-match ($\beta$-sheet).** This task aims to maximize the probability of secondary structure (SS) matching between the generated protein sequences and a predefined target pattern, computed across all residues. SS are predicted using the DSSP (Kabsch & Sander, 1983), and include $\alpha$-helices, $\beta$-sheets, and coils. Following Pacesa et al. (2024), we specifically encourage the formation of $\beta$-sheets, as protein generative models are known to exhibit a bias toward $\alpha$-helices.

- **Binding affinity.** This task focuses on designing binder proteins given target proteins, with the goal of maximizing their binding affinity. We quantify binding affinity using the ipTM score predicted by the AlphaFold-Multimer model (Jumper et al., 2021). Following prior work (Pacesa et al., 2024), we select PD-L1 and IFNAR2 as representative target proteins.

Table 2: Performance of different methods on protein binding design tasks w.r.t. ipTM, optimized reward, and diversity. The best result is highlighted in **bold**.

| Method | PD-L1 | | | IFNAR2 | | |
|---|---|---|---|---|---|---|
| | ipTM↑ | Reward↑ | Diversity↑ | ipTM↑ | Reward↑ | Diversity↑ |
| Pre-trained | $0.1468 \pm 0.0538$ | $0.0847 \pm 0.1317$ | **0.9022** | $0.1179 \pm 0.0153$ | $0.0612 \pm 0.0621$ | 0.9007 |
| Best-of-N (N=128) | $0.2662 \pm 0.1091$ | $0.2654 \pm 0.0629$ | 0.8996 | $0.2463 \pm 0.1055$ | $0.2225 \pm 0.0675$ | 0.9058 |
| Standard Fine-tuning | $0.1640 \pm 0.0215$ | $0.1598 \pm 0.0351$ | 0.8999 | $0.1307 \pm 0.0503$ | $0.0926 \pm 0.0712$ | **0.9063** |
| DDPP | $0.1889 \pm 0.0330$ | $0.2065 \pm 0.0453$ | 0.8763 | $0.1375 \pm 0.0794$ | $0.1236 \pm 0.0782$ | 0.8850 |
| DDPO | $0.7881 \pm 0.0250$ | $0.8767 \pm 0.0301$ | 0.5266 | $0.2403 \pm 0.0488$ | $0.3142 \pm 0.0544$ | 0.7169 |
| VIDD | **$0.8182 \pm 0.0213$** | **$0.9079 \pm 0.0237$** | 0.5539 | **$0.5090 \pm 0.1079$** | **$0.5120 \pm 0.1093$** | 0.5176 |

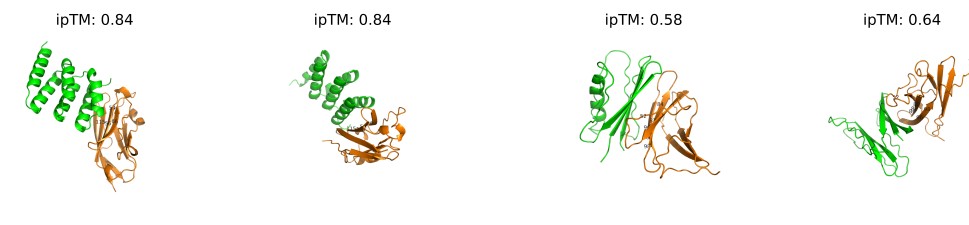

**(a) PD-L1**        **(b) IFNAR2**

Figure 2: Protein structure visualizations for the PD-L1 and IFNAR2 binding design tasks. The binder protein is shown in green and target protein is in orange, with hotspot residues labeled on the structure.

**DNA sequence design.** We focus on the regulatory DNA designs widely used in the cell engineering (Taskiran et al., 2024; Su et al., 2025). Following Wang et al. (2024), we adopt a discrete diffusion model (Sahoo et al., 2024) trained on enhancer datasets from Gosai et al. (2023) as our pre-trained model ($T = 128$). For the reward function, we use predictions from the Enformer model (Avsec et al., 2021) to estimate enhancer activity in the HepG2 cell line (denoted by **Pred-Activity**). This reward has been widely employed in prior DNA design studies (Taskiran et al., 2024; Lal et al., 2024) due to its relevance in cell engineering, particularly for modulating cell differentiation.

**Small molecule design.** We use GDSS (Jo et al., 2022), trained on ZINC-250k (Irwin & Shoichet, 2005), as the pre-trained diffusion model ($T = 1000$). For rewards, we use **binding affinity** to protein Parp1 (Yang et al., 2021) (docking score (**DS**) calculated by QuickVina 2 (Alhossary et al., 2015)), which is non-differentiable. Here, we renormalize docking score to $\max(-\mathrm{DS}, 0)$, so that a higher value indicates better performance. Note these rewards have been widely used Lee et al. (2023); Jo et al. (2022); Yang et al. (2021); Li et al. (2025).

### 6.1.2 BASELINES

We compare **VIDD** with the following baselines.

- **Best-of-N**: This is a naïve yet widely adopted approach for reward maximization at test time. Note that such inference-time methods are significantly slower compared to our fine-tuned models. Additional comparisons will be provided in Appendix E. Given that this method is $N$ times slower than the baseline methods, we do *not* consider it a practical baseline for fine-tuning.

- **Standard Fine-Tuning (SFT).** This method fine-tunes the model by sampling from the pretrained model and applying the same loss used during pretraining, but with reweighting based on reward values (Peng et al., 2019).

- **DDPO (Black et al., 2024)**: A PPO-style algorithm discussed in Section 5.

- **DDPP (Rector-Brooks et al., 2024)**: Another recent state-of-the-art method applicable to our setting with non-differentiable reward functions by enforcing detailed balance between reward-weighted posteriors and denoising trajectories.

- **VIDD**: Our algorithm. Regarding more detailed setting of hyperparameters, refer to Appendix D.2.

### 6.1.3 METRICS

Recall that our objective is to generate samples with high desired reward as in Section 2.2. Accordingly, we report the median reward of generated samples as the primary evaluation metric. As secondary metrics, we include additional measures whose specific choice depends on the task context, such as the naturalness of generated samples. More specifically, we use **pLDDT** scores for protein design; the 3-mer Pearson correlation (**3-mer Corr**) between the generated sequences and those in the dataset from Gosai et al. (2023); the negative log-likelihood (**NLL**) of the generated samples with respect to the pretrained model for small molecule design. For further results, refer to Appendix E.

### 6.2 RESULTS

All results are summarized in Table 1 and Table 2. Below, we provide an interpretation of each outcome.

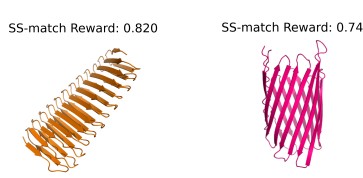

**Protein sequence design.** **VIDD** consistently outperforms baseline methods by achieving higher rewards in $\beta$-sheet content and ipTM binding affinity. Additionally, Figure 2 and Figure 3 illustrate the predicted binder–target complexes and secondary-structure matching tasks, confirming effective binding of the designed binders to their targets and reasonable secondary structures of the designed proteins. Other quantitative results are provided in Appendix E.4, further validating the quality and effectiveness of the generated proteins, as well as effects of varying parameters such as mixture of roll-in policy, lazy update interval and regularization coefficient.

Figure 3: Protein structure visualizations for protein SS-match tasks.

**DNA sequence design.** Here, since this reward is technically differentiable, we include an additional baseline, **DRAKES** (Wang et al., 2024), which directly backpropagates through the reward signal. Notably, as shown in the **Pred-Activity** column, **VIDD** outperforms not only **DDPP** and **DDPO**, but also **DRAKES**. In addition to the aforementioned **Pred-Activity** and **3-mer Corr** metrics, we follow Wang et al. (2024) in incorporating **ATAC-Acc**—an independent binary classification model trained on chromatin accessibility data from the HepG2 cell line (Consortium et al., 2012)—as an orthogonal reward. This is motivated by the fact that **Pred-Activity** is a trained reward model and thus may be susceptible to overoptimization. In the **ATAC-Acc** column, our method also exhibits strong performance, suggesting that **VIDD** would be robust to over-optimization. More quantitative results can be found in Appendix E.2.

**Small molecule design.** **VIDD** outperforms the fine-tuning baseline methods in terms of reward. More quantitative metrics that describe naturalness in molecules is in Appendix E.3.

## 7 CONCLUSION

In this work, we present **VIDD**, a novel fine-tuning algorithm for diffusion models under possibly non-differentiable reward functions. Our method has the potential to accelerate discovery in areas such as protein and drug design, but we also recognize possible misuse in generating harmful biomolecular sequences. We advocate for safeguards and responsible research practices in deploying such models.

ACKNOWLEDGMENTS

This work was supported in part by National Institutes of Health under grant U01AG070112 and ARPA-H under grant 1AY1AX000053.

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

## A    PROOF OF THEOREM 1: CONNECTING PPO OBJECTIVE WITH REVERSE KL DIVERGENCE

We restate the formal version of Theorem 1. The statement is $J(\theta)$ is equal to

$$-\sum_t \mathbb{E}_{\{p_t^\theta\}_{t=T}^0}[\mathrm{KL}(p_{t-1}^\theta(\cdot|x_t)\|p_{t-1}^\star(\cdot|x_t))].$$

up to some constant. We will provide its proof in this section.

Now, we prove this statement. We start calculating the inverse KL divergence. This is

$$\alpha \sum_{t=T+1}^1 \mathrm{KL}\left(p_{t-1}^\theta(\cdot|x_t)\|p_{t-1}^\star(\cdot|x_t)\right)$$

$$=\alpha \sum_{t=T+1}^1 \mathrm{KL}\left(p_{t-1}^\theta(\cdot|x_t)\|\frac{p_{t-1}^{\mathrm{pre}}(\cdot|x_t)\exp(v_{t-1}(\cdot)/\alpha)}{\exp(v_t(x_t)/\alpha)}\right)$$

$$=\alpha \sum_{t=T+1}^1 \mathbb{E}_{\{p_{t-1}^\theta(x_{t-1}|x_t)\}_t}\left[\log p_{t-1}^\theta(x_{t-1}|x_t) + \frac{v_t(x_t)}{\alpha} - \log p_{t-1}^{\mathrm{pre}}(x_{t-1}|x_t) - \frac{v_{t-1}(x_{t-1})}{\alpha}\right]$$

$$=\sum_{t=T+1}^1 \mathbb{E}_{\{p_{t-1}^\theta(x_{t-1}|x_t)\}_t}\left[-r(x_0) + \alpha\mathrm{KL}\left(p_{t-1}^\theta(\cdot|x_t)\|p_{t-1}^{\mathrm{pre}}(\cdot|x_t)\right)\right] + c$$

Here, $c$ corresponds to constant $v_{T+1}(\cdot)$ where it is defined by

$$\alpha \log \mathbb{E}_{x_0 \sim p^{\mathrm{pre}}(x_0|x_T)}[\exp(r(x_0)/\alpha)],$$

recalling its definition (3).

## B    CONNECTING **VIDD** OBJECTIVE WITH KL DIVERGENCE

In this section, we plan to explain how we derive (4). Recall that the original objective is

$$\underset{\theta}{\mathrm{argmin}}\, \mathbb{E}_{x_t \sim u_t}[\mathrm{KL}(p_{t-1}^\star(\cdot|x_t)\|p_{t-1}^\theta(\cdot|x_t))].$$

Based on the definition of conditional KL divergence

$$\mathrm{KL}\big(p(\cdot\mid x_t)\,\|\,q(\cdot\mid x_t)\big) = \mathbb{E}_{x_{t-1}\sim p(\cdot|x_t)}\big[\log p(x_{t-1}\mid x_t) - \log q(x_{t-1}\mid x_t)\big],$$

and the definition of soft optimal policy $p_{t-1}^\star$:

$$p_{t-1}^\star(\cdot|x_t) = \frac{p_{t-1}^{\mathrm{pre}}(\cdot|x_t)\exp(v_{t-1}(\cdot)/\alpha)}{\exp(v_t(x_t)/\alpha)},$$

our objective can be rewritten as

$$\underset{\theta}{\mathrm{argmin}}\, \mathbb{E}_{x_t \sim u_t, x_{t-1}\sim p_{t-1}^\star(x_{t-1}|x_t)}\left[\log p_{t-1}^\star(x_{t-1}\mid x_t) - \log p_{t-1}^\theta(x_{t-1}\mid x_t)\right]$$

$$= \mathrm{argmin}\sum_t \mathbb{E}_{x_t \sim u_t, x_{t-1}\sim p_{t-1}^\star(x_{t-1}|x_t)}[\log p_{t-1}^\star(x_{t-1}|x_t)]$$

$$-\sum_t \mathbb{E}_{x_t \sim u_t}\left[\frac{1}{\exp(v_t(x_t)/\alpha)}\mathbb{E}_{x_{t-1}\sim p_{t-1}^{\mathrm{pre}}(x_{t-1}|x_t)}[\exp(v_{t-1}(x_{t-1})/\alpha)\log p_{t-1}^\theta(x_{t-1}|x_t)]\right].$$

Hence, ignoring the first term since this is constant, the objective function reduces to

$$\underset{\theta}{\mathrm{argmax}}\sum_t \mathbb{E}_{x_t \sim u_t}\left[\frac{1}{\exp(v_t(x_t)/\alpha)}\mathbb{E}_{x_{t-1}\sim p_{t-1}^{\mathrm{pre}}(x_{t-1}|x_t)}[\exp(v_{t-1}(x_{t-1})/\alpha)\log p_{t-1}^\theta(x_{t-1}|x_t)]\right],$$

thus we obtain (5).

## C  BROADER IMPACT AND LIMITATIONS

### C.1  BROADER IMPACT

This paper presents work whose goal is to advance the field of Deep Learning, particularly diffusion models. While this research primarily contributes to technical advancements in generative modeling, it has potential implications in domains such as drug discovery and biomolecular engineering. We acknowledge that generative models, particularly those optimized for specific reward functions, could be misused if not carefully applied. However, our work is intended for general applications, and we emphasize the importance of responsible deployment and alignment with ethical guidelines in generative AI. Overall, our contributions align with the broader goal of machine learning methodologies, and we do not foresee any immediate ethical concerns beyond those generally associated with generative models.

### C.2  LIMITATIONS AND FUTURE WORKS

The success of reward-guided fine-tuning such as reinforcement learning critically depends on the quality of the reward signal. However, in reality, reward functions are often imperfect: they may reflect proxy objectives that are only loosely correlated with real-world biological or chemical utility. Poorly designed rewards can lead to over-optimization and exploit the reward without producing truly meaningful or functional outputs.

Furthermore, post-training is often sensitive to the precision of the reward—when the reward signal is noisy or misaligned, learning can become unstable or entirely fail. In practice, reward evaluation can also be expensive (e.g., involving structure prediction or simulation), making it essential to design reward mechanisms that are not only accurate but also sample-efficient, enabling effective training under limited reward budget.

Finally, since reward signals are imperfect and often expensive to evaluate, hyperparameters that regulate reward exploitation such as rollout frequency, may require some tuning in practice. Overly aggressive configurations can amplify reward noise, while overly conservative choices may cap achievable gains, reflecting a practical tradeoff inherent to reward-driven optimization. Designing fully adaptive schedules for these parameters would require domain-specific assumptions about reward reliability and exploration metrics, which falls outside the scope of this work but represents a promising engineering extension built on top of our framework.

## D  ADDITIONAL DETAILS FOR EXPERIMENT SETTING

### D.1  DETAILS ON TASKS AND REWARD FUNCTIONS

Here we present the detailed descriptions of the task settings and reward functions used in the experiments.

**SS-match.**  The secondary structure matching steers the energy toward user-defined secondary structure. To annotate residue secondary structure, we use the DSSP algorithm (Kabsch & Sander, 1983), which identifies elements such as $\alpha$-helices, $\beta$-sheets and coils. This reward function returns the fraction of residues assigned to the desired secondary structure element. In our case, we aim to maximize the fraction of residues forming $\beta$-sheets, thereby encouraging structures with higher $\beta$-sheet content (Pacesa et al., 2024).

**pLDDT.**  pLDDT (predicted Local Distance Difference Test) is a per-residue confidence score produced by structure prediction models such as AlphaFold and ESMFold. It estimates the local accuracy of predicted atomic positions, with higher scores indicating greater confidence. In protein generation tasks, pLDDT is widely used to evaluate the structural reliability of predicted models, as higher average pLDDT values correlate with well-formed and accurate local geometry. In our setting, we use the average pLDDT score—computed via ESMFold (Lin et al., 2023)—as a reward signal to encourage the generation of structurally confident protein designs.

Table 3: Hyperparameter values for the DNA generation task.

| Hyperparameters | Values |
|---|---|
| Value weight coefficient $\alpha$ | 1.0 |
| Sequence length | 200 |
| # of decoding steps | 128 |
| Update interval K | 5 |
| # of training steps | 2000 |
| Batch size | 32 |
| Learning rate | $1e-4$ |
| Reward | Pred-Activity |

**Protein Binding Affinity.** We aim to generate binder proteins conditioned on given target proteins. To optimize binding affinity, we maximize the ipTM score, a widely used metric calculated by the AlphaFold2-Multimer model (Jumper et al., 2021; Pacesa et al., 2024). To further encourage the naturalness and foldability of the generated binders, we incorporate additional structural confidence terms into the reward function: Reward $=$ ipTM $+ 0.1 \times$ pLDDT $+ 0.02 \times$ radius. The weights are chosen to ensure that ipTM remains the primary optimization objective and is not overwhelmed by the other, more easily optimized components. The pLDDT is computed from the predicted multimer complex using AlphaFold2, while the radius of gyration encourages the formation of well-folded, globular structures (Pacesa et al., 2024).

**Molecule Binding Affinity.** We use the docking program QuickVina 2 (Alhossary et al., 2015) to compute the docking scores following Yang et al. (2021), with exhaustiveness as 1. Note that the docking scores are initially negative values, while we reverse it to be positive and then clip the values to be above 0, *i.e.*. We compute DS regarding protein parp1 (Poly [ADP-ribose] polymerase-1), which is a target protein that has the highest AUROC scores of protein-ligand binding affinities for DUD-E ligands approximated with AutoDock Vina.

**Enhancer HepG2.** We examine a publicly available large dataset on enhancers ($n \approx 700k$) (Gosai et al., 2023), with activity levels measured by massively parallel reporter assays (MPRA) (Inoue et al., 2019), where the expression driven by each sequence is measured. In the Enhancers dataset, each $x$ is a DNA sequence of length 200. We pretrain the masked discrete diffusion model (Sahoo et al., 2024) on all the sequences. We then split the dataset and train two reward oracles (one for finetuning and one for evaluation) on each subset. Each reward oracle is learned using the Enformer architecture (Avsec et al., 2021), while $y \in \mathbb{R}$ is the measured activity in the HepG2 cell line. The Enformer trunk has 7 convolutional layers, each having 1536 channels. as well as 11 transformer layers, with 8 attention heads and a key length of 64. Dropout regularization is applied across the attention mechanism, with an attention dropout rate of 0.05, positional dropout of 0.01, and feedforward dropout of 0.4. The convolutional head for final prediction has 2*1536 input channels and uses average pooling, without an activation function. We learn two These datasets and reward models are widely used in the literature on computational enhancer design (Lal et al., 2024; Sarkar et al., 2024).

## D.2 HYPERPARAMETERS

Here we present the hyperparameters in Table 3, Table 4 and Table 5.

## D.3 SOFTWARE AND HARDWARE

Our implementation is under the architecture of PyTorch (Paszke, 2019). The deployment environments are Ubuntu 20.04 with 48 Intel(R) Xeon(R) Silver, 4214R CPU @ 2.40GHz, 755GB RAM, and graphics cards NVIDIA RTX 2080Ti. All experiments are conducted on a single GPU, selected from NVIDIA RTX 2080Ti, RTX A6000, or NVIDIA H100 with 80GB HBM3 memory, depending on the scale of the task.

Table 4: Hyperparameter values for the protein generation task.

| Hyperparameters | Values |
|---|---|
| Value weight coefficient $\alpha$ | 1.0 |
| Sequence length | 256 (ss-match), 100 (binding affinity) |
| # of decoding tokens each step | 4 |
| Update interval K | 5 (binder IFNAR2), 50 (ss-match, binder PD-L1) |
| # of training steps | 10000 |
| Batch size | 32 |
| Learning rate | $1e-5$ |
| Reward (ss-match) | ss-match |
| Reward (Binding Affinity) | $ipTM + 0.1 \times pLDDT + 0.02 \times radius$ |

Table 5: Hyperparameter values for the molecule generation task.

| Hyperparameters | Values |
|---|---|
| Value weight coefficient $\alpha$ | 6.0 |
| Maximum atom number | 38 |
| # of decoding steps | 1000 |
| Update interval K | 20 |
| # of training steps | 1000 |
| Batch size | 1024 |
| Learning rate | $5e-6$ |
| Reward | Docking Score |

Table 6: Final performance combined with inference-time techniques.

| Method | Protein SS-match | | DNA Enhancer HepG2 | | | Molecule Docking - Parp1 | |
|---|---|---|---|---|---|---|---|
| | $\beta$-sheet%↑ | pLDDT↑ | Pred-Activity↑ | ATAC-Acc↑ | 3-mer Corr↑ | Docking Score↑ | NLL↓ |
| VIDD | $0.83 \pm 0.01$ | $0.82 \pm 0.01$ | $8.28 \pm 0.18$ | $0.820 \pm 0.384$ | 0.162 | $9.4 \pm 1.7$ | $741 \pm 21$ |
| VIDD + BoN (N=32) | $0.84 \pm 0.00$ | $0.82 \pm 0.01$ | $8.40 \pm 0.07$ | $0.750 \pm 0.433$ | 0.152 | $12.1 \pm 1.0$ | $726 \pm 28$ |

### D.4 LICENSES

The dataset for molecular tasks is under Database Contents License (DbCL) v1.0. The pretrained protein generation model EvoDiff is under MIT License. The dataset for DNA task is covered under AGPL-3.0 license. We follow the regulations for all licenses.

## E ADDITIONAL EXPERIMENT RESULTS

### E.1 CONNECTING **VIDD** WITH INFERENCE-TIME TECHNIQUES

As discussed in Section 1.1, inference-time techniques are orthogonal to our approach. Here we clarify the connection between VIDD and inference-time methods. Among various inference-time schemes, the most relevant to VIDD is SVDD (Li et al., 2024). Both methods approximate the value function using

$$v_t(x_t) \approx r(\hat{x}_0(x_t)),$$

but their usage differs fundamentally: SVDD leverages this estimated value to guide the choice of the next denoising step during sampling, whereas VIDD fine-tunes the diffusion model based on the estimated value, leading to a learnable and reusable policy rather than a purely inference-time adjustment.

Furthermore, inference-time techniques can be combined with **VIDD** to further enhance performances. Table 6 and Table 7 presents the results of applying **VIDD** with Best-of-N sampling, demonstrating that additional performance gains can be achieved through this combination.

Table 7: Final performance combined with inference-time techniques for protein binder design.

| Method | PD-L1 | | | | IFNAR2 | | | |
|---|---|---|---|---|---|---|---|---|
| | ipTM↑ | pLDDT↑ | Radius↓ | Diversity↑ | ipTM↑ | pLDDT↑ | Radius↓ | Diversity↑ |
| VIDD | 0.82 ± 0.02 | 0.87 ± 0.04 | -0.12 ± 0.11 | 0.55 | 0.51 ± 0.11 | 0.47 ± 0.05 | 2.20 ± 2.00 | 0.52 |
| VIDD + BoN (N=128) | 0.84 ± 0.00 | 0.91 ± 0.01 | -0.16 ± 0.06 | 0.39 | 0.66 ± 0.02 | 0.55 ± 0.04 | 1.04 ± 0.45 | 0.43 |

Table 8: Performance of different methods on DNA generation tasks. The best result among fine-tuning baselines is highlighted in **bold**, and the second best result is highlighted in underline.

| Method | Pred-Activity↑ | ATAC-Acc↑ | 3-mer Corr↑ | Log-Lik↑ |
|---|---|---|---|---|
| Pre-trained | 0.14 ± 0.26 | 0.000 ± 0.000 | -0.15 | -245 ± 9.1 |
| Best-of-N (N=32) | 1.30 ± 0.64 | 0.000 ± 0.000 | -0.17 | -240 ± 7.2 |
| DRAKES w/o KL | 6.44 ± 0.04 | 0.825 ± 0.028 | 0.307 | -281 ± 0.6 |
| DRAKES | 5.61 ± 0.07 | **0.925 ± 0.006** | **0.887** | -264 ± 0.6 |
| Standard Fine-tuning | 1.17 ± 1.23 | 0.094 ± 0.292 | 0.829 | -263 ± 9.3 |
| DDPP | 5.33 ± 0.94 | 0.305 ± 0.460 | 0.879 | -218 ± 10.4 |
| DDPO | 7.38 ± 0.11 | 0.086 ± 0.280 | 0.398 | **-126 ± 9.5** |
| GLID$^2$E | 7.35 ± 0.07 | 0.906 ± 0.003 | 0.490 | -240 ± 14.2 |
| VIDD | **8.28 ± 0.18** | 0.820 ± 0.384 | 0.162 | -198 ± 8.6 |

## E.2 FURTHER RESULTS FOR DNA GENERATION

**More metrics on DNA generation tasks.** Here we provide additional evaluation results for the DNA generation task in Table 8 to offer a more comprehensive comparison across a broader set of metrics. We add a new baseline GLID$^2$E (Cao et al., 2025) here for comparison. Note that our optimization target is solely the Pred-Activity. Following the prior work (Wang et al., 2024), we fine-tune using one Pred-Activity reward model and evaluate using a different one to avoid data leakage. In terms of performance, VIDD attains strong gains in Pred-Activity, as well as competitive results on ATAC-Acc and Log-Likelihood. Although its 3-mer correlation diverges from that of the pretrained model distribution, the substantially higher functional rewards indicate that VIDD is able to discover novel yet highly effective sequences. This suggests that VIDD explores regions beyond conventional motif statistics while still generating functionally superior designs.

Regarding overall performance, DRAKES benefits from explicit gradient information as discussed in Section 6.1.2. Gradient-based methods are expected to have better performances because they rely on precise token-level gradients, while methods designed for non-differentiable rewards must operate using only sequence-level reward signals. Despite this disadvantage, VIDD still achieves superior performance on the optimized Pred-Activity objective, demonstrating its effectiveness. We include DRAKES in the comparison to help readers better understand the level of results VIDD can achieve.

**Lazy update interval** We study the effect of the lazy update interval $K$ on the DNA sequence design discussed in Section 4.2, and other parameters are provided in Table 3. As shown in Table 9, performance is different under different $K$ and does not improve monotonically with more frequent updates. These results suggest that less updates of roll-out policy can stabilize training and improve optimization.

**Regularization coefficient** We study the effect of the regularization coefficient $\alpha$ as shown in (7). Hyperparameter details for the remaining settings are listed in Table 3 In the DNA sequence design task, the performance comparison is presented in Table 10, show that $\alpha = 1.0$, i.e., keeping the reward distribution unchanged, yields the best performance.

## E.3 FURTHER RESULTS FOR MOLECULE GENERATION

First, we report the diversity comparisons across methods in Table 11 on page 22.

To evaluate the validity of our method in molecule generation, we further report several key metrics that capture different aspects of molecule quality and diversity in Table 12 on page 22.

Table 9: The influence of lazy update interval $K$ on the performances of DNA sequence design.

| Lazy Update Interval | Pred-Activity↑ | ATAC-Acc↑ | 3-mer Corr↑ | Log-Lik↑ |
|---|---|---|---|---|
| 1 | $7.06 \pm 0.35$ | $0.000 \pm 0.000$ | 0.211 | **-156 ± 12.0** |
| 5 | $\mathbf{8.28 \pm 0.18}$ | $\mathbf{0.820 \pm 0.384}$ | 0.162 | $-198 \pm 8.6$ |
| 10 | $7.76 \pm 0.32$ | $0.047 \pm 0.211$ | 0.457 | $-265 \pm 5.5$ |
| 20 | $7.71 \pm 0.37$ | $0.086 \pm 0.280$ | 0.398 | $-265 \pm 5.3$ |
| 50 | $7.23 \pm 0.42$ | $0.484 \pm 0.500$ | **0.470** | $-266 \pm 7.2$ |

Table 10: The influence of regularization coefficient $\alpha$ on the performances of DNA sequence design.

| Regularization Coefficient | Pred-Activity↑ | ATAC-Acc↑ | 3-mer Corr↑ | Log-Lik↑ |
|---|---|---|---|---|
| 0.8 | $8.21 \pm 0.24$ | $0.820 \pm 0.384$ | 0.272 | $-246 \pm 6.2$ |
| 1.0 | $\mathbf{8.28 \pm 0.18}$ | $0.820 \pm 0.384$ | 0.162 | **-198 ± 8.6** |
| 2.0 | $7.26 \pm 0.39$ | $\mathbf{0.977 \pm 0.151}$ | **0.351** | $-248 \pm 8.0$ |

The validity of a molecule indicates its adherence to chemical rules, defined by whether it can be successfully converted to SMILES strings by RDKit. Uniqueness refers to the proportion of generated molecules that are distinct by SMILES string. Novelty measures the percentage of the generated molecules that are not present in the training set. Fréchet ChemNet Distance (FCD) measures the similarity between the generated molecules and the test set. The Similarity to Nearest Neighbors (SNN) metric evaluates how similar the generated molecules are to their nearest neighbors in the test set. Fragment similarity measures the similarity of molecular fragments between generated molecules and the test set. Scaffold similarity assesses the resemblance of the molecular scaffolds in the generated set to those in the test set. The neighborhood subgraph pairwise distance kernel Maximum Mean Discrepancy (NSPDK MMD) quantifies the difference in the distribution of graph substructures between generated molecules and the test set considering node and edge features. Atom stability measures the percentage of atoms with correct bond valencies. Molecule stability measures the fraction of generated molecules that are chemically stable, *i.e.*, whose all atoms have correct bond valencies. Specifically, atom and molecule stability are calculated using conformers generated by RDKit and optimized with UFF (Universal Force Field) and MMFF (Merck Molecular Force Field).

We compare the metrics using 512 molecules generated from the pre-trained GDSS model and from different methods, as shown in Table 12 on page 22. Overall, our method achieves comparable performances with the pre-trained model on all metrics, maintaining high validity, novelty, and uniqueness while outperforming on several metrics such as FCD, SNN, and NSPDK MMD. Pre-trained performs consistently well across all metrics, particularly in SNN and atomic stability. However, it does not optimize specific molecular properties as effectively as the other methods. DDPP performs poorly in scaffold similarity and NSPDK MMD, indicating that it generates unrealistic molecules. These results indicate that our approach can generate a diverse set of novel molecules that are chemically plausible and relevant.

### E.4 FURTHER RESULTS FOR PROTEIN GENERATION

**More metrics on protein binder design**  Tables 13 and Table 14 report the full evaluation metrics for the protein binding affinity design tasks. Here Reward $=$ ipTM $+ 0.1 \times$ pLDDT $+ 0.02 \times$ radius, indicate the aggregated metrics. While ipTM serves as the primary metric for binding affinity, pLDDT and radius of gyration are included as secondary metrics to encourage the generation of well-folded, globular binder proteins.

Additional evaluation metrics are reported in Table 15. We present results for both pTM and pDockQ, where pTM values are obtained directly from AlphaFold2-Multimer (Jumper et al., 2021) and pDockQ scores follow Bryant et al. (2022). From the table, we observe that our proposed VIDD consistently achieves the best performance among the main baselines. Note pTM and pDockQ are not directly optimized during the fine-tuning, but only used for evaluation. Protein binding affinity can be assessed using a wide variety of metrics, raising the question of how to effectively integrate multiple objectives. Developing a principled multi-objective fine-tuning framework therefore remains an interesting direction for future work to explore.

Table 11: Performance of different methods on molecular generation task w.r.t. reward, NLL, and diversity.

| Method | Binding Affinity - Parp1 | | |
|---|---|---|---|
| | Docking Score↑ | NLL↓ | Diversity↑ |
| Pre-trained | 7.2 ± 0.5 | 971 ± 32 | 0.7784 ± 0.2998 |
| Best-of-N (N=32) | 10.2 ± 0.4 | 951 ± 22 | 0.7938 ± 0.3052 |
| Standard Fine-tuning | 7.8 ± 1.8 | 908 ± 77 | 0.8787 ± 0.1088 |
| DDPP | 7.9 ± 1.3 | 981 ± 52 | 0.8067 ± 0.0845 |
| DDPO | 8.5 ± 1.3 | 929 ± 43 | 0.8993 ± 0.0567 |
| VIDD | 9.4 ± 1.7 | 741 ± 21 | 0.9019 ± 0.0477 |
| VIDD + BoN | 12.1 ± 1.0 | 726 ± 28 | 0.9135 ± 0.0509 |

Table 12: Comparison of the generated molecules across various metrics. The best values for each metric are highlighted in **bold**.

| Method | Valid↑ | Unique↑ | Novelty↑ | FCD↓ | SNN↑ | Frag↑ | Scaf↑ | NSPDK MMD↓ | Mol Stable↑ | Atm Stable↑ |
|---|---|---|---|---|---|---|---|---|---|---|
| Pre-trained | **1.000** | **1.000** | **1.000** | 12.979 | 0.414 | 0.513 | **1.000** | 0.038 | 0.320 | 0.917 |
| DPS | **1.000** | **1.000** | **1.000** | 13.230 | 0.389 | 0.388 | **1.000** | 0.040 | 0.310 | 0.878 |
| SMC | **1.000** | 0.406 | **1.000** | 22.710 | 0.225 | 0.068 | **1.000** | 0.285 | 0.000 | **0.968** |
| SVDD | **1.000** | **1.000** | **1.000** | 12.278 | 0.428 | 0.622 | **1.000** | 0.052 | 0.478 | 0.910 |
| Standard Fine-tuning | **1.000** | **1.000** | **1.000** | 9.698 | 0.485 | 0.597 | **1.000** | 0.026 | 0.375 | 0.935 |
| DDPP | **1.000** | 0.671 | **1.000** | 5.280 | 0.426 | **0.759** | NaN | 0.778 | 0.379 | 0.919 |
| DDPO | **1.000** | **1.000** | **1.000** | 11.506 | 0.333 | 0.484 | 0.975 | 0.032 | **0.581** | 0.961 |
| VIDD | **1.000** | **1.000** | **1.000** | **4.869** | **0.489** | 0.714 | 0.999 | **0.016** | 0.458 | 0.911 |

Table 13: Performance of different methods on protein binding design tasks on target protein PD-L1.

| Method | Reward↑ | ipTM↑ | pLDDT↑ | Radius↓ | Diversity↑ |
|---|---|---|---|---|---|
| Pre-trained | 0.0847 ± 0.1317 | 0.1468 ± 0.0538 | 0.3284 ± 0.0420 | 4.7420 ± 5.2352 | **0.9022** |
| Best-of-N (N=128) | 0.2654 ± 0.0629 | 0.2662 ± 0.1091 | 0.3890 ± 0.0530 | 1.9883 ± 3.4164 | 0.8996 |
| Standard Fine-tuning | 0.1598 ± 0.0351 | 0.1640 ± 0.0215 | 0.3349 ± 0.0328 | 1.8829 ± 1.1945 | 0.8999 |
| DDPP | 0.2065 ± 0.0453 | 0.1889 ± 0.0330 | 0.3720 ± 0.0269 | 0.9780 ± 0.9635 | 0.8763 |
| DDPO | 0.8767 ± 0.0301 | 0.7881 ± 0.0250 | 0.8244 ± 0.0821 | **-0.3081 ± 0.1285** | 0.5266 |
| VIDD | **0.9079 ± 0.0237** | **0.8182 ± 0.0213** | **0.8720 ± 0.0421** | -0.1232 ± 0.1066 | 0.5539 |

Table 14: Performance of different methods on protein binding design tasks on target protein IFNAR2.

| Method | Reward↑ | ipTM↑ | pLDDT↑ | Radius↓ | Diversity↑ |
|---|---|---|---|---|---|
| Pre-trained | 0.0612 ± 0.0621 | 0.1179 ± 0.0153 | 0.3525 ± 0.0513 | 4.5964 ± 2.9320 | 0.9007 |
| Best-of-N (N=128) | 0.2225 ± 0.0675 | 0.2463 ± 0.1055 | 0.3843 ± 0.0515 | 3.1082 ± 2.9186 | 0.9058 |
| Standard Fine-tuning | 0.0926 ± 0.0712 | 0.1307 ± 0.0503 | 0.3321 ± 0.0381 | 3.5632 ± 2.1944 | **0.9063** |
| DDPP | 0.1236 ± 0.0782 | 0.1375 ± 0.0794 | 0.3624 ± 0.0359 | 2.5107 ± 1.1723 | 0.8850 |
| DDPO | 0.3142 ± 0.0544 | 0.2403 ± 0.0488 | **0.6300 ± 0.0743** | **-0.5435 ± 0.1219** | 0.7169 |
| VIDD | **0.5120 ± 0.1093** | **0.5090 ± 0.1079** | 0.4711 ± 0.0490 | 2.2039 ± 1.9989 | 0.5176 |

Table 15: Additional evaluation metrics for protein binder design. Note that those metrics are not optimized during training and only used for evaluation.

| Method | PD-L1 | | IFNAR2 | |
|---|---|---|---|---|
| | pTM↑ | pDockQ↑ | pTM↑ | pDockQ↑ |
| DDPP | 0.5537 ± 0.0179 | 0.0693 ± 0.0214 | 0.4697 ± 0.0084 | 0.0359 ± 0.0082 |
| DDPO | 0.7862 ± 0.0809 | 0.2445 ± 0.0478 | 0.5064 ± 0.0178 | 0.0516 ± 0.0190 |
| VIDD | **0.8369 ± 0.0300** | **0.4164 ± 0.0460** | **0.5885 ± 0.0368** | **0.1241 ± 0.0352** |

**Mixed roll-in policy** In Section 4.1, we describe a mixed roll-in strategy that samples trajectories from the pre-trained policy $p_t^{\text{pre}}$ and the roll-out policy $p_t^{\text{out}}$ with probabilities $1 - \beta_s$ and $\beta_s$, respectively. Figure 4 ablates $\beta_s$ and find that injecting a non-zero fraction of $p_t^{\text{pre}}$ (i.e., not always rolling in from $p_t^{\text{out}}$) improves diversity and often yields better overall performance. The specific

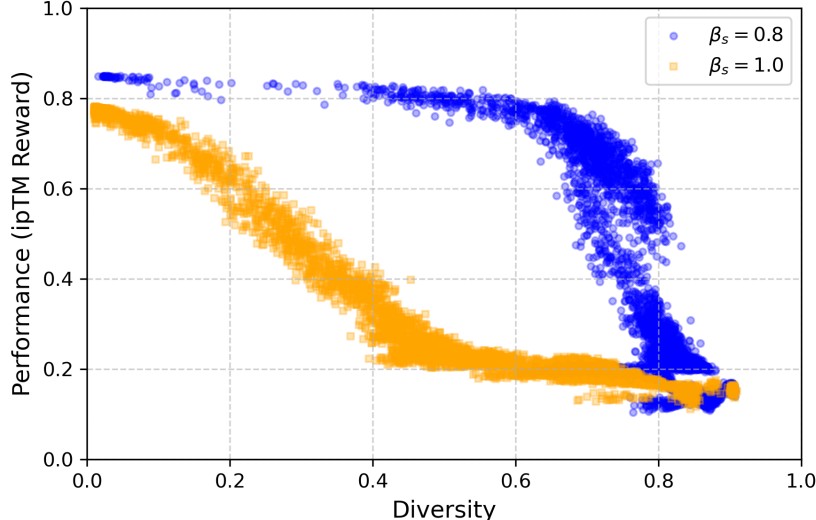

Figure 4: Performance vs. diversity in PD-L1 binder design under different roll-in mixtures. Mixing in the pre-trained policy during roll-in (smaller $\beta_s$) increases diversity compared to relying solely on the roll-out policy ($\beta_s=1$).

Table 16: The influence of lazy update interval $K$ on the performances on protein sequence design for ss-match task.

| Lazy Update Interval $K$ | $\beta$-sheet%↑ | pLDDT↑ | Diversity↑ |
|---|---|---|---|
| 1 | $0.7972 \pm 0.0323$ | $0.6745 \pm 0.0643$ | **0.8238** |
| 5 | **$0.8914 \pm 0.0155$** | $0.6196 \pm 0.0263$ | 0.5023 |
| 50 | $0.8281 \pm 0.0098$ | **$0.8202 \pm 0.0118$** | 0.5154 |

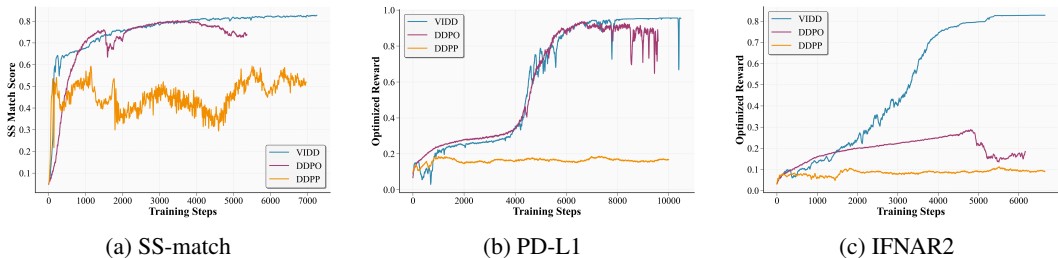

(a) SS-match

(b) PD-L1

(c) IFNAR2

Figure 5: Training curves of different methods on SS-match, PD-L1, and IFNAR2 binder design tasks. The $y$-axis shows the optimized reward, and the $x$-axis shows training steps.

optimal mixture is task-dependent; accordingly, we treat $\beta_s$ as a tunable hyperparameter selected by validation to balance exploration ($p_t^{\text{pre}}$) and exploitation ($p_t^{\text{out}}$).

**Lazy update interval**  For the hyperparameter of lazy update interval $K$ discussed in Section 4.2, we present the results in Table 16, Table 17 and Table 18 for the protein sequence design task, and other parameters are provided in Table 4.

**Regularization coefficient**  For the regularization coefficient $\alpha$ in (7), the performance comparison for protein sequence design tasks is presented in Table 19 and Table 20. We could notice that similar as DNA sequence design, keep the reward distribution unchanged ($\alpha = 1.0$) yields the best performances.

Table 17: The influence of lazy update interval $K$ on the performances on protein binder design tasks for target protein PD-L1.

| Lazy Update Interval $K$ | Reward↑ | ipTM↑ | pLDDT↑ | Radius↓ | Diversity↑ |
|---|---|---|---|---|---|
| 1 | $0.4336 \pm 0.1214$ | $0.4090 \pm 0.1208$ | $0.4174 \pm 0.0304$ | $0.8551 \pm 0.7798$ | 0.5048 |
| 5 | $0.6983 \pm 0.1195$ | $0.6428 \pm 0.1116$ | $0.6211 \pm 0.0814$ | $0.3270 \pm 0.2587$ | 0.5140 |
| 10 | $0.5537 \pm 0.0490$ | $0.5073 \pm 0.0458$ | $0.5606 \pm 0.0374$ | $0.4791 \pm 0.2324$ | 0.5062 |
| 20 | $0.1902 \pm 0.0880$ | $0.2327 \pm 0.0804$ | $0.4367 \pm 0.0678$ | $4.3089 \pm 0.9180$ | 0.5033 |
| 50 | $\mathbf{0.9079 \pm 0.0237}$ | $\mathbf{0.8182 \pm 0.0213}$ | $\mathbf{0.8720 \pm 0.0421}$ | $\mathbf{-0.1232 \pm 0.1066}$ | $\mathbf{0.5539}$ |

Table 18: The influence of lazy update interval $K$ on the performances on protein binder design tasks for target protein IFNAR2.

| Lazy Update Interval $K$ | Reward↑ | ipTM↑ | pLDDT↑ | Radius↓ | Diversity↑ |
|---|---|---|---|---|---|
| 1 | $0.1702 \pm 0.0311$ | $0.1433 \pm 0.0099$ | $0.4195 \pm 0.0321$ | $0.7567 \pm 1.3396$ | 0.7454 |
| 5 | $\mathbf{0.5120 \pm 0.1093}$ | $\mathbf{0.5090 \pm 0.1079}$ | $\mathbf{0.4711 \pm 0.0490}$ | $2.2039 \pm 1.9989$ | 0.5176 |
| 10 | $0.2305 \pm 0.0252$ | $0.1955 \pm 0.0167$ | $0.4517 \pm 0.0302$ | $\mathbf{0.5062 \pm 0.5819}$ | 0.6052 |
| 20 | $0.1528 \pm 0.0352$ | $0.1266 \pm 0.0270$ | $0.3809 \pm 0.0421$ | $0.5971 \pm 0.6893$ | 0.8597 |
| 50 | $0.1227 \pm 0.0231$ | $0.1160 \pm 0.0108$ | $0.3618 \pm 0.0393$ | $1.4747 \pm 0.8167$ | $\mathbf{0.8717}$ |

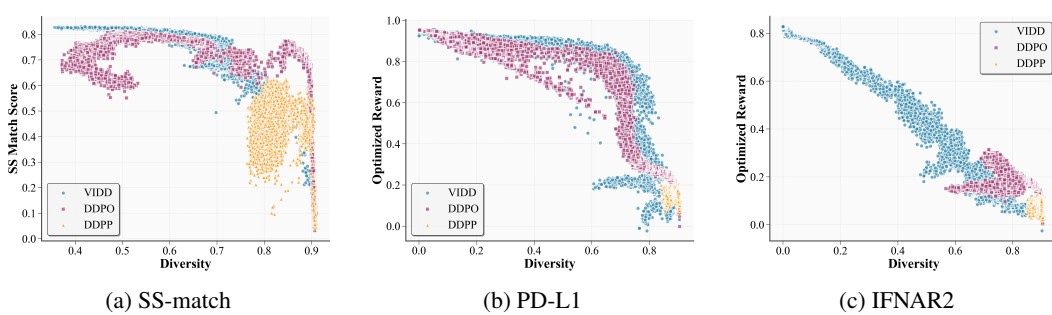

| (a) SS-match | (b) PD-L1 | (c) IFNAR2 |

Figure 6: Scatter plots of reward versus diversity for different methods on SS-match, PD-L1, and IFNAR2 binder design tasks. Each point corresponds to a training checkpoint.

**Trade-off between Reward and Diversity** Readers may be concerned that higher rewards could come at the cost of reduced diversity from results in Table 1 and Table 2 (as well as Table 13 and Table 14). To clarify this, we include the training curves (Figure 5) and the reward–diversity scatter plots (Figure 6). The curves show how reward evolves over training, while the scatter plots visualize the reachable regions in the reward–diversity plane for each method. These figures illustrate that only our method is able to expand the reachable set and achieve substantially higher rewards across training.

## F SOFT VALUE FUNCTION

### F.1 POSTERIOR MEAN APPROXIMATION

We use (3) to define the soft value function:

$$v_t(x) := \alpha \log \mathbb{E}_{x_0 \sim p^{\mathrm{pre}}(x_0 | x_t)} \left[ \exp\left( \frac{r(x_0)}{\alpha} \right) \Big| x_t \right].$$

Recall that the training objective of diffusion models (Section 2.1) is to accurately recover the clean sample $x_0$ from a noisy state $x_t$. In practice, this means that the diffusion model aims to produce a reliable estimate $\hat{x}_0(x_t)$ of the posterior mean $\mathbb{E}[x_0 | x_t]$. Following prior work (Uehara et al., 2025a; Li et al., 2024), substituting this estimate into the expectation above yields the commonly used soft value approximation:

$$v_t(x_t) \approx r(\hat{x}_0(x_t)),$$

which is often referred to as the *posterior mean approximation*. In our implementation, we obtain $\hat{x}_0(x_t)$ via argmax decoding from the distribution $p(x_0 \mid x_t)$ at $t = 0$.

Table 19: The influence of regularization coefficient $\alpha$ on the performances on protein binder design tasks for target protein PD-L1.

| Regularization Coefficient $\alpha$ | Reward↑ | ipTM↑ | pLDDT↑ | Radius↓ | Diversity↑ |
|---|---|---|---|---|---|
| 0.8 | $0.5505 \pm 0.0809$ | $0.4839 \pm 0.0739$ | $0.5430 \pm 0.0565$ | $\mathbf{-0.6149 \pm 0.1902}$ | $\mathbf{0.5592}$ |
| 1 | $\mathbf{0.9079 \pm 0.0237}$ | $\mathbf{0.8182 \pm 0.0213}$ | $\mathbf{0.8720 \pm 0.0421}$ | $-0.1232 \pm 0.1066$ | $0.5539$ |
| 2 | $0.4443 \pm 0.0854$ | $0.3930 \pm 0.0812$ | $0.4752 \pm 0.0552$ | $-0.1865 \pm 0.3129$ | $0.5038$ |

Table 20: The influence of regularization coefficient $\alpha$ on the performances on protein binder design tasks for target protein IFNAR2.

| Regularization Coefficient $\alpha$ | Reward↑ | ipTM↑ | pLDDT↑ | Radius↓ | Diversity↑ |
|---|---|---|---|---|---|
| 0.8 | $0.2729 \pm 0.0844$ | $0.2472 \pm 0.0733$ | $0.4013 \pm 0.0448$ | $\mathbf{0.7226 \pm 0.6606}$ | $0.5017$ |
| 1 | $\mathbf{0.5120 \pm 0.1093}$ | $\mathbf{0.5090 \pm 0.1079}$ | $\mathbf{0.4711 \pm 0.0490}$ | $2.2039 \pm 1.9989$ | $0.5176$ |
| 2 | $0.1005 \pm 0.0480$ | $0.1247 \pm 0.0455$ | $0.3587 \pm 0.0329$ | $3.0035 \pm 1.3196$ | $\mathbf{0.8769}$ |

### F.2 MONTE CARLO ESTIMATION

Beyond the posterior mean, one may also approximate the soft value using Monte Carlo sampling, which computes multiple predictions of $x_0$ and averages their rewards:

$$v_t(x_t) \approx \frac{1}{M} \sum_{m=1}^{M} r\left(\hat{x}_0^{(m)}(x_t)\right), \qquad \hat{x}_0^{(m)}(x_t) \sim p(x_0 \mid x_t),$$

where we sample from $p(x_0 \mid x_t)$ multiple times to get different $\hat{x}_0^{(m)}(x_t)$ by temperature sampling. Temperature sampling is used because argmax decoding above produces a single deterministic mode and therefore cannot support multiple samples for Monte Carlo estimation.

Table 21 and Table 22 reports the effect of different soft value estimation functions in our setting. Specifically, Posterior mean represents take the argmax decoding to get $x_0$ from $p(x_0 \mid x_t)$. The interesting observation is that even when we use Monte Carlo estimation with $M = 4$, which increases reward computation cost by $4\times$ (and reward calculation itself is expensive using models like ESMFold–3B or AlphaFold2-Multimer-93M parameters), the performance becomes worse. This suggests that the posterior mean (argmax) provides a more stable and discriminative value signal for estimating $v_t(x_t)$. In contrast, small-$M$ Monte Carlo samples mix high- and low-quality draws, weakening the training guidance. Increasing $M$ further could reduce this variance, but it would require substantially more reward calls, making it computationally impractical in biomolecular design, where reward evaluation is both high-cost and non-regular.

Therefore, using argmax decoding to approximate the value offers the best trade-off: it minimizes reward computation while providing a stable and reliable signal for fine-tuning, making it the most practical choice for VIDD.

### F.3 TRAINING VALUE NETWORK

Another alternative is to train a separate neural network to approximate the soft value function directly. Although such critic models are common in standard reinforcement learning, they are considerably less practical in biomolecular design. Reward functions in our domains (e.g., protein binder design, secondary-structure–matching design) rely on large structure prediction models such as AlphaFold2-Multimer (Jumper et al., 2021) (93.2M parameters) or ESMFold (Lin et al., 2023) (3B parameters), which operate on full sequences rather than individual denoising steps. In contrast, the diffusion generator used in our experiments (EvoDiff (Alamdari et al., 2023)) contains only 38M parameters. Training an accurate value network over intermediate states $x_t$ instead of the final state is extremely challenging and computationally extensive. Consequently, we focus on the posterior mean and Monte Carlo estimators in this work, and leave the training of scalable and accurate value networks for complex biomolecular tasks as an interesting future direction.

Table 21: The influences of soft value function on the performances on protein binder design tasks for target protein PD-L1.

|  | Reward↑ | ipTM↑ | pLDDT↑ | Radius↓ | Diversity↑ |
|---|---|---|---|---|---|
| Posterior mean | **0.9079 ± 0.0237** | **0.8182 ± 0.0213** | **0.8720 ± 0.0421** | -0.1232 ± 0.1066 | **0.5539** |
| Monte carlo estimation (M=4) | 0.7758 ± 0.0620 | 0.7105 ± 0.0603 | 0.4993 ± 0.0359 | **-0.7711 ± 0.1049** | 0.5155 |

Table 22: The influences of soft value function on the performances on protein binder design tasks for target protein IFNAR2.

|  | Reward↑ | ipTM↑ | pLDDT↑ | Radius↓ | Diversity↑ |
|---|---|---|---|---|---|
| Posterior mean | **0.5120 ± 0.1093** | **0.5090 ± 0.1079** | 0.4711 ± 0.0490 | 2.2039 ± 1.9989 | 0.5176 |
| Monte carlo estimation (M=4) | 0.4149 ± 0.0936 | 0.3474 ± 0.0906 | **0.6892 ± 0.0617** | **0.0703 ± 0.2881** | **0.5195** |

## G  NOISE REWARD FUNCTION

In this section, we inject synthetic noise into the reward estimation function to assess the robustness of VIDD under imperfect reward signals. This scenario reflects realistic biomolecular settings, where surrogate reward models may deviate from true experimental measurements. Specifically, we add $n\%$ Gaussian noise to the reward function as:

$$r_n = r + r \cdot n\% \cdot \mathcal{N}(0,1),$$

where $\mathcal{N}(0,1)$ is the normal distribution. During training, we use the noised reward $r_n$ as signal, and for inference, we observe the clean reward $r$. The results in Table 23 and Table 24 show that, as noise levels increase, the performance degrades noticeably. This is expected: VIDD does not incorporate explicit robustness mechanisms, so inaccuracies in reward estimation can mislead the optimization process and fine-tune the model toward suboptimal directions.

Since noisy or biased rewards exist in biomolecular design, addressing reward uncertainty remains a crucial direction. Incorporating robustness-aware reinforcement learning techniques, such as reward denoising or uncertainty calibration, represents a promising direction for future research.

## H  VISUALIZATION OF GENERATED SAMPLES

In Figure 7 we visualizes the docking of **VIDD** generated molecular ligands to protein parp1. Docking scores presented above each column quantify the binding affinity of the ligand-protein interaction, while the figures include various representations and perspectives of the ligand-protein complexes. We aim to provide a complete picture of how each ligand is situated within both the local binding environment and the larger structural framework of the protein. First rows show close-up views of the ligand bound to the protein surface, displaying the topography and electrostatic properties of the protein's binding pocket and providing insight into the complementarity between the ligand and the pocket's surface. Second rows display distant views of the protein using the surface representation, offering a broader perspective on the ligand's spatial orientation within the global protein structure. Third rows provide close-up views of the ligand interaction using a ribbon diagram, which represents the protein's secondary structure, such as alpha-helices and beta-sheets, to highlight the specific regions of the protein involved in binding. Fourth rows show distant views of the entire protein structure in ribbon diagram, with ligands displayed within the context of the protein's full tertiary structure. Ligands generally fit snugly within the protein pocket, as evidenced by the close-up views in both the surface and ribbon diagrams, which show minimal steric clashes and strong surface complementarity.

Table 23: The influences of noise reward function on the performances on protein binder design tasks for target protein PD-L1.

|  | Reward↑ | ipTM↑ | pLDDT↑ | Radius↓ | Diversity↑ |
|---|---|---|---|---|---|
| noise=0.0 | **0.9079 ± 0.0237** | **0.8182 ± 0.0213** | **0.8720 ± 0.0421** | **-0.1232 ± 0.1066** | **0.5539** |
| noise=0.1 | 0.4744 ± 0.1172 | 0.4570 ± 0.1192 | 0.4306 ± 0.0255 | 1.2802 ± 0.9287 | 0.5154 |

Table 24: The influences of noise reward function on the performances on protein binder design tasks for target protein IFNAR2.

|  | Reward↑ | ipTM↑ | pLDDT↑ | Radius↓ | Diversity↑ |
|---|---|---|---|---|---|
| noise=0.0 | **0.5120 ± 0.1093** | **0.5090 ± 0.1079** | **0.4711 ± 0.0490** | 2.2039 ± 1.9989 | 0.5176 |
| noise=0.1 | 0.1970 ± 0.0217 | 0.1446 ± 0.0160 | 0.4692 ± 0.0620 | **-0.2763 ± 0.2740** | **0.5234** |

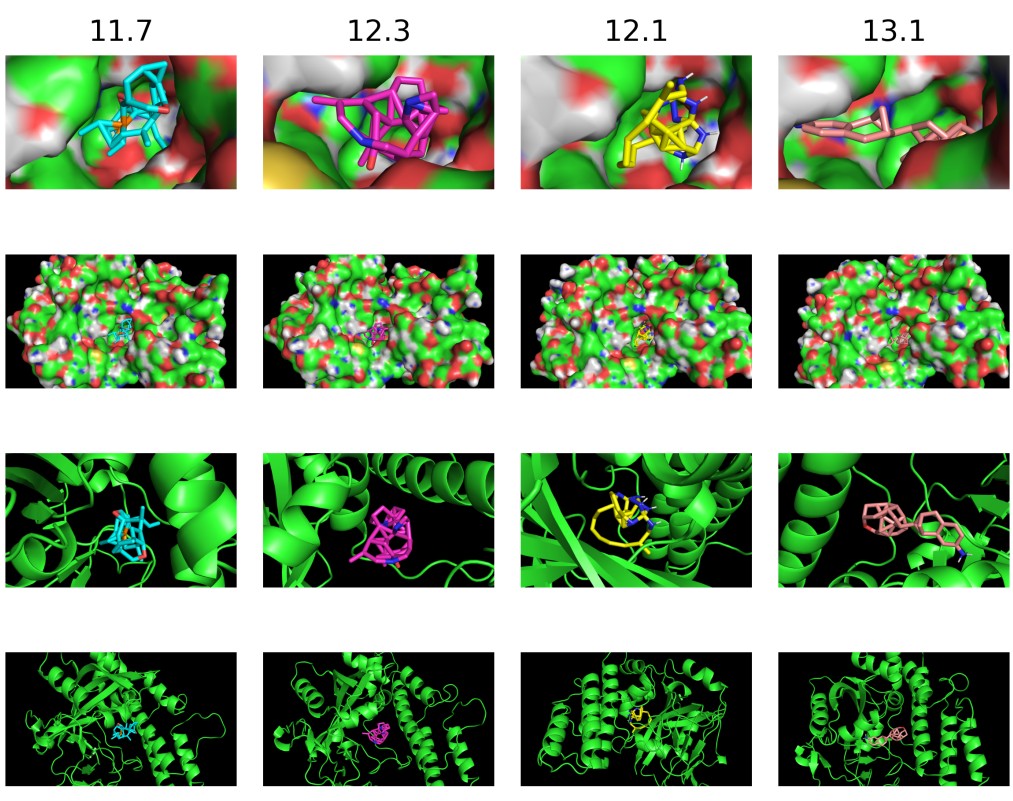

Figure 7: Visualization of generated molecules using **VIDD** optimizing the reward of docking score for parp1 (normalized as $max(-DS, 0)$).

