# OpenReview forum: "Iterative Distillation for Reward-Guided Fine-Tuning of Diffusion Models in Biomolecular Design"
_ICLR.cc/2026/Conference — ICLR 2026 Poster_

### Official Review · Reviewer_AJss · 2025-10-30

**Soundness:** 3
**Presentation:** 3
**Contribution:** 3
**Rating:** 6
**Confidence:** 4

**Summary:**

The paper

**Strengths:**

In general, the exposition is well done. I don't have a ton of experience on modern RL approaches for training diffusion models and I found the explanations fairly approachable.

The evaluation setup is from prior work from papers that are more application-specific. In general, creating reward functions that are cheap computational surrogates for wet-lab experiments is difficult. I work in this area. I think the ones that are used are reasonable choices.

**Weaknesses:**

It was hard for me to understand which parts of Section 4 are novel. See question below.

The main paper provides few experimental results. However, the supplement has some really interesting follow-on results. It would have been nice to see some of these ablations and analyses featured in the main paper (e.g., fig 4).

The 'approximate soft-value functions' approach makes approximations that may be risky to make in practice. I would have liked to see a contrast with a sampling-based approach to estimating this.

**Questions:**

I found the explanation of 'Approximation of soft value functions' (line 8 in Alg 1) a little too concise. In what sense is this a posterior mean? Why is it possible to evaluate a discrete reward function on this 'mean'?

It was hard for me to understand which parts of Section 4 are novel. Can you please provide more details about the relationship to prior work? Also, are there special cases or modifications of your framework that would make the method similar to other RL methods (e.g. if the roll-out policy was the same as the roll-in policy)? Drawing these connections can help readers understand things better. I appreciate section 6, but it was hard for me to understand if there is other related work that is similar to yours instead of a policy gradient method.

To what extent is the use of a diffusion model orthogonal to the particular RL approach? The inverse process of the diffusion model is just just a generic policy. Why is your presentation specific to diffusion models? Are there other non-diffusion policies where your approach would be sensible?

What if, for example, you applied your approach to an autoregressive protein language model instead of a discrete diffusion model? I'm not suggesting you do this experiment. I'm just trying to understand why diffusion models are the focal point.

---

> ### Author Response · Authors · 2025-11-23
>
> Thanks for the valuable feedback and insightful comments. We address each concern in detail below, and the corresponding revisions have been updated in the manuscript. All changes are highlighted in blue italics for clarity.
>
> > The main paper provides few experimental results. However, the supplement has some really interesting follow-on results. It would have been nice to see some of these ablations and analyses featured in the main paper (e.g., fig 4).
>
> Thanks for the feedback. Due to the strict page limit, we prioritized presenting the core contributions and main results in the primary manuscript, and therefore some of the ablations (e.g., Fig. 4) were moved to the appendix. We now explicitly reference these analyses from the main paper to improve visibility. If accepted, we will re-organize the results into the camera-ready version, where an additional page would allow us to include more of these insightful results directly in the main text.
>
> > The 'approximate soft-value functions' approach makes approximations that may be risky to make in practice. I would have liked to see a contrast with a sampling-based approach to estimating this.
>
> > I found the explanation of 'Approximation of soft value functions' (line 8 in Alg 1) a little too concise. In what sense is this a posterior mean? Why is it possible to evaluate a discrete reward function on this 'mean'?
>
>
> Thanks for raising this point. Our approximation leverages a key property of diffusion models: they are trained to recover the clean sample $x_0$​ from a noisy intermediate state $x_t$. In practice, the model provides a reliable estimate $\hat{x}_0(x_t)$, which approximates the posterior mean $\mathbb{E}[x_0 \mid x_t]$. This leads to our value approximation $v_t(x_t) \approx r(\hat{x}_0(x_t))$, referred to as the posterior mean approximation. A similar idea has also been explored in prior work (SVDD-PM [1]).
>
> In implementation, we obtain $\hat{x}_0(x_t)$ via argmax decoding from the distribution $p(x_0 \mid x_t)$. Alternatively, we can approximate the soft value by Monte Carlo sampling, computing multiple predictions of $x_0$​ and averaging their rewards. We compare both strategies below, with further details provided in Appendix F and in Tables 21 and 22.
>
> PD-L1 binder design
> |                       | Reward↑              | ipTM↑               | pLDDT↑              | Radius↓                 | Diversity↑ |
> |-----------------------|----------------------|---------------------|---------------------|--------------------------|-------------|
> | Posterior mean    | **0.9079 ± 0.0237**  | **0.8182 ± 0.0213** | **0.8720 ± 0.0421** | -0.1232 ± 0.1066     | **0.5539**  |
> | Monte carlo estimation (M=4) | 0.7758 ± 0.0620  | 0.7105 ± 0.0603     | 0.4993 ± 0.0359     | **-0.7711 ± 0.1049**         | 0.5155      |
>
>
> IFNAR2 binder design
> |                       | Reward↑              | ipTM↑               | pLDDT↑              | Radius↓                 | Diversity↑ |
> |-----------------------|----------------------|---------------------|---------------------|--------------------------|-------------|
> | Posterior mean    | **0.5120 ± 0.1093**  | **0.5090 ± 0.1079** | 0.4711 ± 0.0490 | 2.2039 ± 1.9989      | 0.5176  |
> | Monte carlo estimation (M=4) | 0.4149 ± 0.0936  | 0.3474 ± 0.0906     | **0.6892 ± 0.0617**     | **0.0703 ± 0.2881**          | **0.5195**      |
>
>
> We can observe that posterior mean estimation yields higher reward performance compared with Monte Carlo estimator, even Monte Carlo estimator will have N times computation on reward calculation. This suggests that the posterior mean (argmax) provides a more stable and discriminative value signal for estimating $v_t(x_t)$. Given that biomolecular tasks often require a high-cost reward model (e.g., ESMFold with 3B parameters, AlphaFold2-Multimer with 93M parameters, whereas the generative model adopted is EvoDiff with 38M parameters), we choose the posterior-mean decoding to approximate the value.

---

> ### Author Response · Authors · 2025-11-23
>
> > It was hard for me to understand which parts of Section 4 are novel. Can you please provide more details about the relationship to prior work? Also, are there special cases or modifications of your framework that would make the method similar to other RL methods (e.g. if the roll-out policy was the same as the roll-in policy)? Drawing these connections can help readers understand things better. I appreciate section 6, but it was hard for me to understand if there is other related work that is similar to yours instead of a policy gradient method.
>
> Thanks for raising this point. The novelty of VIDD lies in how it leverages the diffusion trajectory rather than how rewards are weighted. Eq.(2) and Eq.(3) show that VIDD defines a soft value at intermediate states $x_t$​ to guide the step-wise fine-tuning, instead of assigning reward only to the final outcome $x_0$ to optimize the whole trajectory​ as in prior reward-weighted MLE methods. Section 4 shows how to make this theoretical formulation practical: Section 4.1 proposes mixture roll-in to control diversity, and Section 4.2 provides practical strategies to approximate the soft value, enabling the method to work in real biomolecular systems where reward gradients are not accessible. These components make VIDD work in a diffusion-compatible manner. To further clarify the connection to existing RL methods, VIDD contains two limiting cases:
>
>  (i) If we remove the intermediate value and only consider the path from initial noise $x_T$ to the final sample $x_0$​, the method reduces to reward-weighted MLE.
>
>  (ii) If both the roll-in and roll-out policies are fixed to the pretrained generator, VIDD becomes offline fine-tuning using only logged samples, similar to offline RL.
>
> We hope such clarifications could help readers to understand VIDD better.
>
> > To what extent is the use of a diffusion model orthogonal to the particular RL approach? The inverse process of the diffusion model is just just a generic policy. Why is your presentation specific to diffusion models? Are there other non-diffusion policies where your approach would be sensible?
>
> > What if, for example, you applied your approach to an autoregressive protein language model instead of a discrete diffusion model? I'm not suggesting you do this experiment. I'm just trying to understand why diffusion models are the focal point.
>
>
> Thanks for the helpful question. VIDD combines with diffusion models well on the approximation of soft value functions. Recall diffusion models are trained to denoise $x_t$​ back to a clean sample $x_0$​. This allows the model’s prediction $\hat{x}_0(x_t)$ to serve as a practical proxy for the conditional expectation $\mathbb{E}[x_0 \mid x_t]$, leading to the value approximation $v_t(x_t) \approx r(\hat{x}_0(x_t))$. This enables value-guided training without an explicit value network or differentiable rewards. Such an estimator does not generally exist for arbitrary policies, whose intermediate states lack a principled connection to a clean $x_0$​. This is why our framework is diffusion models. A clarification has been added in Appendix F.
>
> In this way, this kind of soft value estimation doesn’t quite fit autoregressive models. In diffusion models, every intermediate state $x_t$​ already encodes a denoised approximation of the final clean sample, because the model is explicitly trained to map noisy states back toward $x_0$​. In contrast, for autoregressive language models, an intermediate prefix only predicts the next token distribution, not a meaningful approximation of the complete final sequence. Such models do not provide an estimate of the full $x_0$​, making it difficult to compute value functions at intermediate steps. Therefore, while our framework conceptually applies beyond diffusion, diffusion uniquely provides the structural property required for tractable soft value estimation.
>
>
>
>
>
>
> [1] Li X, Zhao Y, Wang C, et al. Derivative-free guidance in continuous and discrete diffusion models with soft value-based decoding[J]. arXiv preprint arXiv:2408.08252, 2024.

---

> > ### Comment · Reviewer_AJss · 2025-11-25
> >
> > Thanks for the comprehensive response. I have raised my review to 'Accept. I found your response to the 'To what extent is the use of a diffusion model orthogonal to the particular RL approach?' question important. Please make sure to update the manuscript to make this argument clear to the reader.

---

> > > ### Author Response · Authors · 2025-11-25
> > >
> > > Thank you for taking the time to consider the reviews and for raising the score.
> > > We will update the paper to clearly show the connection between diffusion models and RL approach.
> > > Thank you again for your thoughtful feedbacks.

---

### Official Review · Reviewer_F9Zw · 2025-10-30

**Soundness:** 3
**Presentation:** 2
**Contribution:** 2
**Rating:** 4
**Confidence:** 4

**Summary:**

This paper focuses on addressing the challenge of reward-guided fine-tuning of diffusion models in biomolecular design. Diffusion models are effective at modeling the complex distributions of biomolecules like proteins, small molecules, and regulatory DNA. However, real-world applications require optimizing for core rewards—such as physics-based protein binding affinity, small molecule docking scores, and science-driven protein secondary structure matching scores—that are non-differentiable, making traditional gradient-based methods inapplicable. Existing reinforcement learning (RL)-based fine-tuning approaches (e.g., DDPO, PPO) face issues including training instability, low sample efficiency, and mode collapse due to their on-policy nature and reliance on reverse KL divergence.

To tackle these problems, the authors propose VIDD (Value-guided Iterative Distillation for Diffusion models), a framework that treats fine-tuning as a policy distillation task, enabling optimization for any type of reward (including non-differentiable ones) through three iterative stages. First, the Roll-in stage collects off-policy trajectories by mixing samples from the pre-trained diffusion model (to explore a wide range of biomolecular design spaces) and a stable roll-out policy (to leverage high-reward regions), decoupling data collection from policy updates to improve sample efficiency. Second, the Roll-out stage simulates a soft-optimal "teacher policy" that balances reward maximization and compliance with the pre-trained distribution. This policy is weighted by a value function approximated using the diffusion model’s prediction of the clean sample , eliminating the need for reward gradients. Third, the Distillation stage updates the "student model" (the fine-tuned diffusion model) by minimizing the forward KL divergence between the soft-optimal policy and the current model policy, which prevents mode collapse compared to the reverse KL divergence used in RL methods.

Experiments across four biomolecular design tasks—protein secondary structure matching, protein-target binding, DNA enhancer design, and small molecule docking—demonstrate that VIDD consistently outperforms baseline methods (e.g., DDPO, DDPP, Best-of-N) in terms of reward metrics. For instance, it achieves 83% β-sheet content in proteins, 8.28 Pred-Activity in DNA enhancers, and a 9.4 docking score for small molecules. Meanwhile, it maintains biomolecular naturalness, as shown by metrics like pLDDT (a measure of protein structure confidence) and 3-mer correlation (a measure of DNA sequence naturalness).

**Strengths:**

1.Effective Adaptation to Non-Differentiable Rewards: VIDD bypasses the need for reward gradients by approximating soft value functions using the diffusion model’s x₀ prediction. This allows it to directly optimize core non-differentiable rewards in biomolecular design, such as binding affinity scores from AlphaFold and docking scores from QuickVina2, filling a key gap in fine-tuning diffusion models for scientific applications.
2. Training Stability and Anti-Collapse Ability: Unlike on-policy RL methods, VIDD uses off-policy data in the Roll-in stage, reducing sensitivity to noisy trajectories. By minimizing forward KL divergence instead of the reverse KL divergence used in RL, it forces the model to cover the distribution of the soft-optimal policy, avoiding mode collapse. For example, in protein binding tasks, VIDD maintains higher diversity than DDPO, and in small molecule design, it achieves a lower negative log-likelihood (NLL), indicating better preservation of biomolecular naturalness.

**Weaknesses:**

1.	Unaddressed Impact of Value Function Approximation Errors: VIDD relies on the diffusion model’s x₀prediction to approximate soft value functions, but it does not analyze how errors in x₀prediction—such as those arising from long protein sequences or complex DNA structures —affect the distillation process. Additionally, it fails to compare this approximation method with alternatives like Monte Carlo sampling, leaving uncertainty about whether this is the optimal approach.
2.	High Hyperparameter Sensitivity and Lack of Adaptive Strategies: Key hyperparameters—such as the roll-in mix ratio (βₛ), the roll-out policy update interval (K), and the value weight coefficient (α)—require manual tuning and have a significant impact on performance. For example, in DNA tasks, using K = 5 results in much higher activity than using K = 20. However, VIDD does not propose adaptive strategies (e.g., dynamically adjusting K based on reward progress), which increases the difficulty of applying it across different tasks.
3.	Insufficient Exploration of Robustness to Noisy Rewards: Biomolecular rewards often suffer from "proxy bias"—for instance, a predicted binding affinity score (ipTM) only approximates real binding affinity, and DNA activity predicted by Enformer may differ from actual cellular activity. VIDD does not test its performance under conditions of reward noise (e.g., random fluctuations) or bias, leaving unanswered questions about whether it will over-optimize for proxy objectives (e.g., producing proteins with high predicted binding affinity but no actual binding function).

**Questions:**

1.	Value Function Approximation and Error Correlation: VIDD uses x₀ prediction to approximate soft value functions, but the accuracy of x₀ prediction tends to decrease for long biomolecules (e.g., large protein complexes or DNA sequences with over 500 bases). Can you provide additional experiments to show the correlation between x₀ prediction errors (e.g., structural similarity between the predicted x₀ and the real x₀) and VIDD’s reward performance? Or can you compare this approximation method with alternatives such as Monte Carlo sampling or regression-based value estimation to verify its validity?
2.	Adaptive Hyperparameter Optimization: Hyperparameters like the Roll-in mix ratio (βₛ) and Roll-out update interval (K) have a major impact on performance. Manual tuning is time-consuming—for example, βₛ=0.8 optimizes protein diversity. Do you plan to design adaptive strategies, such as adjusting K based on the rate of reward improvement or optimizing βₛ based on the diversity of Roll-in data? Can you also provide heuristic rules for hyperparameter selection across different biomolecular tasks (e.g., recommended ranges for α in protein vs. small molecule tasks)?
3.	Robustness to Noisy and Biased Rewards: Biomolecular rewards often contain noise—for example, fluctuations in Enformer-predicted DNA activity or small molecule docking scores. How would VIDD perform if rewards are perturbed (e.g., ±10% Gaussian noise) or biased (e.g., systematically underestimating high-reward samples)? Are there calibration methods (e.g., multi-reward fusion) to reduce the impact of proxy bias?

---

> ### Author Response · Authors · 2025-11-23
>
> Thanks for the valuable feedback and insightful comments. We address each concern in detail below, and the corresponding revisions have been updated in the manuscript. All changes are highlighted in blue italics for clarity.
>
> >W1. Unaddressed Impact of Value Function Approximation Errors: VIDD relies on the diffusion model’s x₀ prediction to approximate soft value functions, but it does not analyze how errors in x₀ prediction—such as those arising from long protein sequences or complex DNA structures —affect the distillation process. Additionally, it fails to compare this approximation method with alternatives like Monte Carlo sampling, leaving uncertainty about whether this is the optimal approach.
>
> >Q1. Value Function Approximation and Error Correlation: VIDD uses x₀ prediction to approximate soft value functions, but the accuracy of x₀ prediction tends to decrease for long biomolecules (e.g., large protein complexes or DNA sequences with over 500 bases). Can you provide additional experiments to show the correlation between x₀ prediction errors (e.g., structural similarity between the predicted x₀ and the real x₀) and VIDD’s reward performance? Or can you compare this approximation method with alternatives such as Monte Carlo sampling or regression-based value estimation to verify its validity?
>
>
>
> Thanks for pointing it out. Quantifying the prediction error of the value estimator is not well-defined because there is no ground-truth $x_0$. Given an intermediate state $x_t$, there may exist multiple plausible clean samples consistent with the diffusion transition. Therefore, rather than recovering one true $x_0$, the goal is to approximate the expectation over all valid $x_0$, which is infeasible to enumerate.
>
> Our approximation leverages a key property of diffusion models: they are explicitly trained to recover a clean sample $x_0$ from a noisy state $x_t$. In practice, the model provides a reliable estimate $\hat{x}_0(x_t)$ of the posterior mean $ \mathbb{E}[x_0 \mid x_t] $, which reduces to our approximation $v_t(x_t) \approx r(\hat{x}_0(x_t))$. A similar idea is also presented in previous works (SVDD-PM [1]).
>
> Besides, we include a comparison between our posterior-mean decoding (argmax) and temperature-based Monte Carlo estimation (see Tables below, more details are in Table 21 and Table 22 in Appendix F). We can observe that posterior mean estimation yields higher reward performance compared with Monte Carlo estimator, even Monte Carlo estimator will have N times computation on reward calculation. This suggests that the posterior mean (argmax) provides a more stable and discriminative value signal for estimating $v_t(x_t)$. Given that biomolecular tasks often require a high-cost reward model (e.g., ESMFold with 3B parameters, AlphaFold2-Multimer with 93M parameters, whereas the generative model adopted is EvoDiff with 38M parameters), we choose the posterior-mean decoding to approximate the value.
>
> PD-L1 binder design
> |                       | Reward↑              | ipTM↑               | pLDDT↑              | Radius↓                 | Diversity↑ |
> |-----------------------|----------------------|---------------------|---------------------|--------------------------|-------------|
> | Posterior mean    | **0.9079 ± 0.0237**  | **0.8182 ± 0.0213** | **0.8720 ± 0.0421** | -0.1232 ± 0.1066     | **0.5539**  |
> | Monte carlo estimation (M=4) | 0.7758 ± 0.0620  | 0.7105 ± 0.0603     | 0.4993 ± 0.0359     | **-0.7711 ± 0.1049**         | 0.5155      |
>
>
> IFNAR2 binder design
> |                       | Reward↑              | ipTM↑               | pLDDT↑              | Radius↓                 | Diversity↑ |
> |-----------------------|----------------------|---------------------|---------------------|--------------------------|-------------|
> | Posterior mean    | **0.5120 ± 0.1093**  | **0.5090 ± 0.1079** | 0.4711 ± 0.0490 | 2.2039 ± 1.9989      | 0.5176  |
> | Monte carlo estimation (M=4) | 0.4149 ± 0.0936  | 0.3474 ± 0.0906     | **0.6892 ± 0.0617**     | **0.0703 ± 0.2881**          | **0.5195**      |
>
>
>
>
> Another approach is directly learning a value network as in standard RL, which is less practical in biomolecular design. As we discussed, reward computation requires large structure prediction models. Training a value model over intermediate states $x_t$ is extremely challenging and computationally extensive. Hence, our method focuses on diffusion-based value approximation rather than explicit value learning. We provide further analysis and additional ablations in Appendix F for clarity.

---

> ### Author Response · Authors · 2025-11-23
>
> >W2. High Hyperparameter Sensitivity and Lack of Adaptive Strategies: Key hyperparameters—such as the roll-in mix ratio (βₛ), the roll-out policy update interval (K), and the value weight coefficient (α)—require manual tuning and have a significant impact on performance. For example, in DNA tasks, using K = 5 results in much higher activity than using K = 20. However, VIDD does not propose adaptive strategies (e.g., dynamically adjusting K based on reward progress), which increases the difficulty of applying it across different tasks.
>
> >Q2. Adaptive Hyperparameter Optimization: Hyperparameters like the Roll-in mix ratio (βₛ) and Roll-out update interval (K) have a major impact on performance. Manual tuning is time-consuming—for example, βₛ=0.8 optimizes protein diversity. Do you plan to design adaptive strategies, such as adjusting K based on the rate of reward improvement or optimizing βₛ based on the diversity of Roll-in data? Can you also provide heuristic rules for hyperparameter selection across different biomolecular tasks (e.g., recommended ranges for α in protein vs. small molecule tasks)?
>
>
> Thanks for the insightful question. We agree that VIDD introduces several hyperparameters, including the roll-in mixture ratio $ \beta_s $, the rollout update interval $K$, and the value weight $ \alpha $. Below we clarify their roles and practical guidelines.
>
> - $\alpha$: this is the most straightforward to tune. Across all experiments (DNA and protein tasks; Table 10, 19, 20), $ \alpha = 1.0 $ consistently performs best because it preserves the original reward distribution without re-scaling. Hence, we recommend simply setting $ \alpha = 1.0 $, avoiding additional tuning effort.
>
> - $\beta_s$: this parameter controls the balance between pretrained diversity and reward-seeking exploration. A similar design appears in concurrent work GLID$^2$E (Section 4.3 and Table 5 of [2]). Empirically, readers could start with $ \beta_s = 1.0 $ and observe the diversity/naturalness score. If diversity/naturalness shows significantly decreases during training, decreasing $\beta_s$ to introduce more pretrained roll-in. However, fully relying on pretrained roll-in ($\beta_s=0$), is often suboptimal (also consistent with [2]).
> - $K$: this parameter stabilizes optimization by preventing overly frequent policy changes. Its role is analogous to the target-network update interval in DQN [3], where lazy updates are known to improve stability. Here, we present different results under different $K$ in Table 9, 16, 17, 18. We observe that $K=1$ is not a good option. Our recommendation is that users can gradually increase the value of $K$ to find the optimal parameters.
>
> Overall speaking, the rollout update interval $K$ is more difficult to be searched for, while the other two are generally easy to obtain. From the results in Table 9, 10, 16, 17, 18, 19, 20, users can observe how different values influence stability and reward improvement, which offers a more clear guideline for selecting hyperparameters in practice.
>
> Finally, although adaptive schedules for $K$ are conceptually appealing, applying them in biomolecular design would require task-specific assumptions about reward distribution, diversity metrics, or update triggers, which may differ greatly across protein and DNA settings, since those reward functions (e.g., AlphaFold, ESMFold) are complex. Introducing such mechanisms may add new hyperparameters, shifting the focus away from our core objective: demonstrating that value-based distillation alone can fine-tune diffusion models without datasets or differentiable supervision. We therefore treat adaptive rollout control as a practical engineering extension, complementary but not essential to the scientific contribution of this work. We have added this discussion in the limitation section in the revised version (Appendix C.2).

---

> ### Author Response · Authors · 2025-11-23
>
> >W3. Insufficient Exploration of Robustness to Noisy Rewards: Biomolecular rewards often suffer from "proxy bias"—for instance, a predicted binding affinity score (ipTM) only approximates real binding affinity, and DNA activity predicted by Enformer may differ from actual cellular activity. VIDD does not test its performance under conditions of reward noise (e.g., random fluctuations) or bias, leaving unanswered questions about whether it will over-optimize for proxy objectives (e.g., producing proteins with high predicted binding affinity but no actual binding function).
>
> >Q3. Robustness to Noisy and Biased Rewards: Biomolecular rewards often contain noise—for example, fluctuations in Enformer-predicted DNA activity or small molecule docking scores. How would VIDD perform if rewards are perturbed (e.g., ±10% Gaussian noise) or biased (e.g., systematically underestimating high-reward samples)? Are there calibration methods (e.g., multi-reward fusion) to reduce the impact of proxy bias?
>
>
>
> We thank the reviewer for raising this important and broadly relevant question. Reward imperfection in biomolecular design is indeed a field-level challenge. Two aspects are involved here:
>
> (1) proxy rewards may not perfectly correlate with real biological functionality;
>
> (2) noisy or biased reward signals may adversely affect learning.
>
> On the first point, (i) In practice, one may design multiple reward signals, e.g., for protein binder design, we combine ipTM, pLDDT, and radius of gyration. However, how to choose, calibrate, and weight such objectives is a reward-modeling problem, not a generator training problem. Multi-reward optimization itself is an active research direction in biomolecular design. (ii) Within our scope, we report additional unoptimized metrics (pTM and pDockQ in Table 15), and visualize generated structures to confirm that VIDD does not simply generate trivial sequences/structures, but produces biologically meaningful sequences under multiple measurements. (iii) Ultimately, resolving proxy bias requires experimentally validated reward systems, since computational metrics are proxies by definition. Wet-lab validation is necessary to determine whether an objective truly correlates with biological function, and this lies outside the scope of generative fine-tuning methods, including ours.
>
>
>
>
> For the second point, regarding noise robustness, we now include experiments where we inject Gaussian noise (±10%) into the reward function (details are in Appendix G).
>
>
>
> PD-L1
> |               | Reward↑              | ipTM↑               | pLDDT↑              | Radius↓                | Diversity↑ |
> |---------------|----------------------|---------------------|---------------------|-------------------------|-------------|
> | noise=0.0 | **0.9079 ± 0.0237**  | **0.8182 ± 0.0213** | **0.8720 ± 0.0421** | **-0.1232 ± 0.1066**    | **0.5539**  |
> | noise=0.1     | 0.4744 ± 0.1172      | 0.4570 ± 0.1192     | 0.4306 ± 0.0255     | 1.2802 ± 0.9287         | 0.5154      |
>
> IFNAR2
> |               | Reward↑              | ipTM↑               | pLDDT↑              | Radius↓                | Diversity↑ |
> |---------------|----------------------|---------------------|---------------------|-------------------------|-------------|
> | noise=0.0 | **0.5120 ± 0.1093**  | **0.5090 ± 0.1079** | **0.4711 ± 0.0490** | 2.2039 ± 1.9989     | 0.5176  |
> | noise=0.1     | 0.1970 ± 0.0217      | 0.1446 ± 0.0160     | 0.4692 ± 0.0620     | **-0.2763 ± 0.2740**        | **0.5234**  |
>
>
> As expected, the performance degrades when the reward becomes unreliable, showing that the fine-tuning behavior depends directly on the quality of the reward model (discussed in Appendix C.2). This is an expected outcome for reward-driven optimization methods. Improving robustness would therefore require not only modifying the generator, but also designing better reward models, potentially through multi-objective calibration or uncertainty-aware scoring. However, such strategies require domain-dependent assumptions about the distributions of rewards or how different biological rewards should be weighted (e.g., binding affinity vs. structural stability vs. expression), which remain open challenges in the whole biomolecular design area. These reward modeling problems are independent of how the generator absorbs reward guidance; VIDD focuses on the latter, providing an efficient way to distill reward signals into diffusion models.
>
>
>
>
> [1] Li X, Zhao Y, Wang C, et al. Derivative-free guidance in continuous and discrete diffusion models with soft value-based decoding[J]. arXiv preprint arXiv:2408.08252, 2024.
>
> [2] https://openreview.net/pdf?id=AHjspi4R22.
>
> [3] https://www.nature.com/articles/nature14236

---

### Official Review · Reviewer_dS5g · 2025-10-30

**Soundness:** 2
**Presentation:** 3
**Contribution:** 2
**Rating:** 4
**Confidence:** 4

**Summary:**

The paper proposed VIDD, a novel reinforcement-learning-based method for diffusion models in biomolecule design, including protein, small molecule and DNA sequecnces. The authors found the limitations of the original methods, including mode collapse, computational inefficiency, and reward hacking during training. Then, VIDD solves them step by step with corresponding techniques. In experiments, results from three systems demonstrated the effectiveness of the general design.

**Strengths:**

1. The motivation and the insight for the method design make sense, and the proposed solutions are simple and easy-to-implement.

2. It involves many biological systems, demonstrating the effectiveness and the robustness of the algorithm in different tasks.

3. VIDD unfied diffusion and value-weighted MLE with clear objective function and implementation, the framework could combine any type of non-differentiable reward functions, which is of high application range in biomolecule design.

**Weaknesses:**

1. The most concerning point from me is that the papre did not prove the effectiveness of each component through ablation studies, which makes the Method section not solid. More results on different biological systems are needed.

2. Only limited baseline works were discussed and compared. Some baselines from discrete flow matching/diffusion on biological sequence design are of high correlation with this work [1-4].

3. More strict theoretical proof/analysis are needed (See Questions).

4. There are irrational design of experimental validation (See Questions).

References:

[1] Tang, S., Zhang, Y., & Chatterjee, P. PepTune: De Novo Generation of Therapeutic Peptides with Multi-Objective-Guided Discrete Diffusion. In Forty-second International Conference on Machine Learning.

[2] Zhao, Y., Uehara, M., Scalia, G., Kung, S., Biancalani, T., Levine, S., & Hajiramezanali, E. Adding Conditional Control to Diffusion Models with Reinforcement Learning. In The Thirteenth International Conference on Learning Representations.

[3] Cao, H., Shi, H., Wang, C., Pan, S. J., & Heng, P. A. (2025). GLID $^ 2$ E: A Gradient-Free Lightweight Fine-tune Approach for Discrete Sequence Design. In ICLR 2025 Workshop on Generative and Experimental Perspectives for Biomolecular Design.

[4] Tang, S., Zhang, Y., Tong, A., & Chatterjee, P. (2025). Gumbel-softmax flow matching with straight-through guidance for controllable biological sequence generation. ArXiv, arXiv-2503.

**Questions:**

1. The results in DNA design system of DRAKES is different from the ones from original paper. Is there any setting changes?

2. In Section 5, the paper presents a theorem showing that the PPO-style objective is equivalent to a reverse KL divergence to the soft optimal trajectory distribution. However, for VIDD, it only states that it is “closer to a forward KL,” without providing a theorem or assumptions under which the empirical objective (7) is strictly equivalent to a forward KL.

3. Eq (5) and Eq(7) fall under the offline-RL framework. Could you please clarify the coverage condition for hte roll-in distribution. Moreover, can you analyze the coveragence under mixture roll-ins?

4. Can you do ablation studies on K, $\alpha$, to demonstrate the stability if lazy updates?

5. Also for Eq (5) and Eq (7), what is the differences/novelty of VIDD compared to exising reward-reweighted SFT or value-weighted MLE in offline RL?

6.For the “off-policy + forward-KL”, is there empirical evidence showing that VIDD maintains its claimed stability advantage under the same roll-in coverage and identical KL regularization budget?

7. Can you provide a formal proof/connection between VIDD and the mentioned inference-time methods?

8. I found that when the roll-in distribution and target teacher distribution are mismatched, Eq. (7) does not perform explicit importance correction. Will it bring biased estimation? If so, how is there any sols?

9. There should be general (mean/median) evaluation for sampling rather than simply using Best-Of-N.

10. More ablation studies should be added.

11. Using AlphaFold-Multimer’s ipTM as the sole metric can be biased by the predictor itself. Can you try to report another metric to demonstrate the effectiveness, such as pTM, pDockQ?

12. For protein experiments, I think there would be potential data mismatch between EvoDiff and VIDD. Would you please clarify it?

13. Could you please show some insights towards the training distillation and test-time sampling tradeoff?

14. Can you also report log-likelihood in DNA system? It is because sometimes the 3-mer correlation cannot capture accurate simiarity among distributions.

---

> ### Author Response · Authors · 2025-11-23
>
> We sincerely thank the reviewers for their valuable feedback and insightful comments. We address each concern in detail below, and the corresponding revisions have been updated in the manuscript. All changes are highlighted in blue italics for clarity.
>
> > Q1. The results in DNA design system of DRAKES is different from the ones from original paper. Is there any setting changes?
>
> > Q14. Can you also report log-likelihood in DNA system? It is because sometimes the 3-mer correlation cannot capture accurate similarity among distributions.
>
>
> Thank you for the question.
>
> - We confirm that there are **no setting changes** to the original experimental settings of DRAKES.  DRAKES provides two versions in the official implementation: with and without the KL regularization term. Since our main objective is to optimize Pred-Activity, we report the version that achieves higher optimized reward. For completeness, we provide both variants in the revised table below and add it into Appendix E.2.
> - We also add the metrics of log-likelihood in the table below (more details are in Table 8 in revised paper).
> - We need to point out that the DRAKES results are shown mainly to provide readers with a reference for the performance level that can be achieved when reward gradients are available. Unlike DRAKES, our method (VIDD) does **not** require access to reward gradients, making it applicable to settings where the reward function is non-differentiable or its gradients are expensive or impractical to compute. Besides, although gradient-based optimization may in principle achieve stronger performance, VIDD still attains higher Pred-Activity, highlighting its practical effectiveness under this more general setting.
>
> |  | Pred-Activity↑  | ATAC-Acc↑ | 3-mer Corr↑ | Log-Lik (median) ↑ |
> | :---- | :---- | :---- | :---- | :---- |
> | DRAKES w/o KL | 6.44 ± 0.04  | 0.825 ± 0.028 | 0.307 | \-281 ± 0.6 |
> | DRAKES | 5.61 ± 0.07  | 0.925 ± 0.006 | 0.887 | \-264 ± 0.6 |
> | DDPP | 5.33 ± 0.94  | 0.305 ± 0.460 | 0.879 | \-218 ± 10.4 |
> | DDPO | 7.38 ± 0.11  | 0.086 ± 0.280 | 0.398 | \-126 ± 9.5 |
> | VIDD | 8.28 ± 0.18  | 0.820 ± 0.384 | 0.162 | \-198 ± 8.6 |
>
>
> > Q2. In Section 5, the paper presents a theorem showing that the PPO-style objective is equivalent to a reverse KL divergence to the soft optimal trajectory distribution. However, for VIDD, it only states that it is “closer to a forward KL,” without providing a theorem or assumptions under which the empirical objective (7) is strictly equivalent to a forward KL.
>
> Thank you for raising this point. We prove that (5) and (7) are equivalent to a forward KL objective in Appendix B, and have revised the appendix to improve readability. Appendix B now provides a clearer step-by-step derivation from the forward KL divergence to (5). To make this connection more transparent, we also explicitly reference the full proof in Section 3.2.
>
> > Q3. Eq (5) and Eq(7) fall under the offline-RL framework. Could you please clarify the coverage condition for hte roll-in distribution. Moreover, can you analyze the coveragence under mixture roll-ins?
>
> Thank you for the thoughtful question. The concern about “coverage conditions” typically applies to offline RL, where learning is restricted to a fixed dataset with limited support. In contrast, our method does **not** rely on a pre-collected dataset. VIDD queries rewards directly from a pretrained reward model (e.g., AlphaFold, ESMFold), allowing every generated sequence to be evaluated. As a result, there is no coverage limitation on the roll-in distribution.
>
> To avoid confusion, we have clarified in the paper that VIDD is off-policy, but not an offline RL method. We apologize for the earlier confusion in the paper, and we also updated the terminology in the revision. Since rewards can be computed for all sampled sequences, using mixture roll-in policies does not introduce convergence issues related to coverage.

---

> ### Author Response · Authors · 2025-11-23
>
> >W1. The most concerning point from me is that the papre did not prove the effectiveness of each component through ablation studies, which makes the Method section not solid. More results on different biological systems are needed.
>
> >Q4. Can you do ablation studies on K, $\alpha$, to demonstrate the stability if lazy updates?
>
> >Q10. More ablation studies should be added.
>
> Thanks for pointing this out. We add additional ablation studies, including the effect of the lazy update interval K and the regularization coefficient $\alpha$ across different tasks. The results are provided below and also added in Appendix E.2 and Appendix E.4 (Table 9, 10, 16, 17, 18, 19, 20). Due to time constraints during the rebuttal, we report the some representative parameters here, and we will include further results and broader hyperparameter explorations in the future camera-ready version.
>
> After observing the results, we find those parameters have influences on the final performances. For $\alpha$, $\alpha = 1.0$ yields the best results, as it preserves the original reward distribution without distortion. For the lazy update interval $K$, tuning is beneficial: $K=1$ always tends to underperform due to overly frequent updates, and we recommend starting around $K \approx 5$ and increasing gradually if needed.
>
>
> K on the DNA generation task
> | Lazy Update Interval | Pred-Activity ↑     | ATAC-Acc ↑         | 3-mer Corr ↑ | Log-Lik ↑        |
> |----------------------|---------------------|---------------------|--------------|------------------|
> | 1                    | 7.06 ± 0.35         | 0.000 ± 0.000       | 0.211        | **-156 ± 12.0**  |
> | 5                    | **8.28 ± 0.18**     | **0.820 ± 0.384**   | 0.162        | -198 ± 8.6       |
> | 10                   | 7.76 ± 0.32         | 0.047 ± 0.211       | 0.457        | -265 ± 5.5       |
> | 20                   | 7.71 ± 0.37         | 0.086 ± 0.280       | 0.398        | -265 ± 5.3       |
> | 50                   | 7.23 ± 0.42         | 0.484 ± 0.500       | **0.470**    | -266 ± 7.2       |
>
> K on protein sequence design of ss-match
> | Lazy Update Interval K | $\beta$-sheet%↑              | pLDDT↑              | Diversity↑ |
> |--------------------------|------------------------|---------------------|-------------|
> | 1                        | 0.7972 ± 0.0323        | 0.6745 ± 0.0643     | **0.8238**  |
> | **5**                    | **0.8914 ± 0.0155**    | 0.6196 ± 0.0263     | 0.5023      |
> | 50                       | 0.8281 ± 0.0098        | **0.8202 ± 0.0118** | 0.5154      |
>
>
> K on the protein PD-L1 binder design
> | Lazy Update Interval K | Reward ↑           | ipTM ↑             | pLDDT ↑          | Radius ↓            | Diversity ↑      |
> |------------------------|--------------------|--------------------|------------------|---------------------|------------------|
> | 1                      | 0.4336 ± 0.1214    | 0.4090 ± 0.1208    | 0.4174 ± 0.0304  | 0.8551 ± 0.7798     | 0.5048           |
> | 5                      | 0.6983 ± 0.1195    | 0.6428 ± 0.1116    | 0.6211 ± 0.0814  | 0.3270 ± 0.2587     | 0.5140           |
> | 10                     | 0.5537 ± 0.0490    | 0.5073 ± 0.0458    | 0.5606 ± 0.0374  | 0.4791 ± 0.2324     | 0.5062           |
> | 20                     | 0.1902 ± 0.0880    | 0.2327 ± 0.0804    | 0.4367 ± 0.0678  | 4.3089 ± 0.9180     | 0.5033           |
> | 50                     | **0.9079 ± 0.0237**| **0.8182 ± 0.0213**| **0.8720 ± 0.0421**| **-0.1232 ± 0.1066** | **0.5539**       |
>
> K on the protein IFNAR2 binder design
> | Lazy Update Interval K | Reward ↑           | ipTM ↑             | pLDDT ↑          | Radius ↓            | Diversity ↑      |
> |------------------------|--------------------|--------------------|------------------|---------------------|------------------|
> | 1                      | 0.1702 ± 0.0311    | 0.1433 ± 0.0099    | 0.4195 ± 0.0321  | 0.7567 ± 1.3396     | 0.7454           |
> | 5                      | **0.5120 ± 0.1093**| **0.5090 ± 0.1079**| **0.4711 ± 0.0490**| 2.2039 ± 1.9989     | 0.5176           |
> | 10                     | 0.2305 ± 0.0252    | 0.1955 ± 0.0167    | 0.4517 ± 0.0302  | **0.5062 ± 0.5819** | 0.6052           |
> | 20                     | 0.1528 ± 0.0352    | 0.1266 ± 0.0270    | 0.3809 ± 0.0441  | 0.5971 ± 0.6893     | 0.8597           |
> | 50                     | 0.1227 ± 0.0231    | 0.1160 ± 0.0108    | 0.3618 ± 0.0393  | 1.4747 ± 0.8167     | **0.8717**       |

---

> ### Author Response · Authors · 2025-11-23
>
> $\alpha$ on the DNA generation task
> | Regularization Coefficient | Pred-Activity ↑     | ATAC-Acc ↑         | 3-mer Corr ↑ | Log-Lik ↑        |
> |----------------------------|---------------------|---------------------|--------------|------------------|
> | 0.8                        | 8.21 ± 0.24         | 0.820 ± 0.384       | 0.272        | -246 ± 6.2       |
> | 1.0                        | **8.28 ± 0.18**     | 0.820 ± 0.384       | 0.162        | **-198 ± 8.6**   |
> | 2.0                        | 7.26 ± 0.39         | **0.977 ± 0.151**   | **0.351**    | -248 ± 8.0       |
>
> $\alpha$ on the protein PD-L1 binder design
> | Regularization Coefficient α | Reward ↑           | ipTM ↑             | pLDDT ↑          | Radius ↓            | Diversity ↑      |
> |------------------------------|--------------------|--------------------|------------------|---------------------|------------------|
> | 0.8                          | 0.5505 ± 0.0809    | 0.4839 ± 0.0739    | 0.5430 ± 0.0565  | **-0.6149 ± 0.1902**| **0.5592**       |
> | 1                            | **0.9079 ± 0.0237**| **0.8182 ± 0.0213**| **0.8720 ± 0.0421**| -0.1232 ± 0.1066    | 0.5539           |
> | 2                            | 0.4443 ± 0.0854    | 0.3930 ± 0.0812    | 0.4752 ± 0.0552  | -0.1865 ± 0.3129    | 0.5038           |
>
> $\alpha$ on the protein IFNAR2 binder design
> | Regularization Coefficient α | Reward ↑           | ipTM ↑             | pLDDT ↑          | Radius ↓            | Diversity ↑      |
> |------------------------------|--------------------|--------------------|------------------|---------------------|------------------|
> | 0.8                          | 0.2729 ± 0.0844    | 0.2472 ± 0.0733    | 0.4013 ± 0.0448  | **0.7226 ± 0.6606** | 0.5017           |
> | 1                            | **0.5120 ± 0.1093**| **0.5090 ± 0.1079**| **0.4711 ± 0.0490**| 2.2039 ± 1.9989     | 0.5176           |
> | 2                            | 0.1005 ± 0.0480    | 0.1247 ± 0.0455    | 0.3587 ± 0.0329  | 3.0035 ± 1.3196     | **0.8769**       |

---

> ### Author Response · Authors · 2025-11-23
>
> >Q5. Also for Eq (5) and Eq (7), what is the differences/novelty of VIDD compared to exising reward-reweighted SFT or value-weighted MLE in offline RL?
>
> Thank you for the question. Existing reward-weighted objectives are typically defined only on the final generated sample $x_0$​ (as in Eq.(1)), and thus operate on the entire trajectory at once. In contrast, VIDD optimizes each intermediate step through the approximate soft value $v_t$​ at time $t$. Instead of performing RL only at the terminal sample, we advocate fine-tuning directly over the soft value function (Eq.(3)), which provides a step-wise, granular and diffusion-compatible training signal.
>
> This formulation leverages a key property of diffusion models: they are trained to recover the clean sample $x_0$​ from noisy states $x_t$​. As a result, the model’s prediction $ \hat{x}_0(x_t) $ serves as a practical proxy for the conditional expectation $ \mathbb{E}[x_0 \mid x_t] $, leading to the approximation $ v_t(x_t) \approx r(\hat{x}_0(x_t)) $. Furthermore, there exist multiple ways to approximate $v_t$​. In our main experiments, we adopt a posterior mean approximation, and we add a more detailed discussion and comparison of different soft-value estimation strategies (e.g., Monte Carlo estimation) in Appendix F.
>
> >Q6. For the “off-policy + forward-KL”, is there empirical evidence showing that VIDD maintains its claimed stability advantage under the same roll-in coverage and identical KL regularization budget?
>
> Thank you for the question. To clarify the comparison, we can compare it with “on-policy + reverse KL”. Note that switching from forward KL to reverse KL fundamentally alters both the sampling distributions and the resulting objective. Specifically:
>
>  (1) Reverse KL requires sampling from the current policy $p_{\theta}$​, which forces the method to operate on-policy. As a consequence, the roll-out design discussed in Section 4.2 no longer applies, since sampling must be updated continuously to track the current policy.
>
>  (2) Reverse KL also yields a totally different optimization objective (see Appendix A of our paper and Section 3.3 of DDPP [2]).
>
> Given these differences, we can compare VIDD directly against DDPP (with the same budget and roll-in), which represents a reverse-KL alternative under its proper objective and sampling distributions. The empirical results (Figure 5 in Appendix E.4) show that VIDD consistently achieves significantly better performance, particularly in the ss-match task, whereas DDPP is highly unstable. While numerical “stability” is difficult to measure in isolation, the overall learning curves demonstrate that VIDD improves reliably throughout training, whereas DDPP frequently collapses. This provides practical evidence that combining “off-policy + forward-KL” objective yields more robust optimization behavior in our design tasks.
>
>
> >Q7. Can you provide a formal proof/connection between VIDD and the mentioned inference-time methods?
>
> >Q13. Could you please show some insights towards the training distillation and test-time sampling tradeoff?
>
> Thanks for your questions. VIDD and inference-time optimization share a common goal: given a pretrained unconditional generative model and a reward function, both aim to generate samples that maximize the reward. The key difference lies in where reward guidance is applied. For example, both VIDD and SVDD [1] estimates the intermediate value estimation; however, VIDD directly fine-tunes the model, updates model parameters, and the reward orientation becomes part of the model itself. In contrast, SVDD keeps the pretrained model fixed and only applies reward guidance to select denoising steps during sampling, without modifying the parameters (more discussions are in Appendix E.1).
>
> This distinction leads to different tradeoffs. When the pretrained model cannot be modified (e.g., due to API access or limited training resources), inference-time search becomes preferable. Conversely, when inference efficiency matters, distillation is advantageous because reward-oriented behavior is embedded into the model, eliminating the need for costly reward queries during sampling.
>
> Importantly, the two approaches are not mutually exclusive. As shown in Appendix E.1, combining training distillation (VIDD) with inference-time sampling strategies can further improve reward-oriented generation. In this work, we evaluate a simple Best-Of-N combination, and we view more sophisticated hybrid strategies as an interesting direction for future research.

---

> ### Author Response · Authors · 2025-11-23
>
> >Q8. I found that when the roll-in distribution and target teacher distribution are mismatched, Eq. (7) does not perform explicit importance correction. Will it bring biased estimation? If so, how is there any sols?
>
> Thank you for raising this point. We believe the concern originates from a misunderstanding of the role of the roll-in distribution in Eq.(7). Starting from Eq.(4), the derivation shows that the roll-in distribution $u_t$​ can be arbitrary, and therefore mismatching between $u_t$​ and the teacher distribution does not introduce bias. The requirement instead lies on the roll-out distribution, which must correspond to the pretrained model distribution when computing Eq.(7). Our implementation follows this requirement.
>
> In practice, we periodically update the roll-out policy during fine-tuning. However, this update is performed in a lazy manner, meaning the policy remains nearly unchanged over a sequence of updates and can be treated as a new “pretrained” policy for subsequent steps. As a result, the importance correction does not change the theoretical derivation, and no bias is introduced.
>
> >Q9. There should be general (mean/median) evaluation for sampling rather than simply using Best-Of-N.
>
> Thank you for the helpful suggestion. If the intention is to report the mean performance of sampling-based methods, this is already reflected for the “Pretrained” model, as it directly samples from the original generative model and reports the averaged evaluation. Regarding the Best-of-N results, the reported scores are also computed over a batch of generated samples, and are therefore mean values rather than single-sample statistics.
>
> We would also like to clarify that Best-of-N is shown purely as an inference-time reference, rather than as a baseline for comparison with VIDD. It illustrates the performance one might obtain if the reward model could be queried repeatedly during sampling, without any fine-tuning. Our intention is to show that VIDD can achieve strong performance without requiring such expensive reward queries at test time.
>
> >Q11. Using AlphaFold-Multimer’s ipTM as the sole metric can be biased by the predictor itself. Can you try to report another metric to demonstrate the effectiveness, such as pTM, pDockQ?
>
> Thanks a lot for your suggestions. We report pTM and pDockQ scores for protein binder design tasks in the tables below (also in Table 15 in Appendix E.4). We can observe that VIDD still outperforms the baselines in pTM and pDockQ scores. Here pTM and pDockQ are not used during the fine-tuning, but only for evaluation. Note that protein binding affinity can be assessed using many different metrics, raising the question of how to combine different objectives. Developing multi-objective fine-tuning frameworks remains an interesting direction for future work to explore.
>
>
> | PD-L1 | pTM | pDockQ |
> | :---- | :---- | :---- |
> | DDPP | 0.5537 ± 0.0179 | 0.0693 ± 0.0214 |
> | DDPO | 0.7862 ± 0.0809 | 0.2445 ± 0.0478 |
> | VIDD | **0.8369 ± 0.0300** | **0.4164 ± 0.0460** |
>
> | IFNAR2 | pTM | pDockQ |
> | :---- | :---- | :---- |
> | DDPP | 0.4697 ± 0.0084 | 0.0359 ± 0.0082 |
> | DDPO | 0.5064 ± 0.0178 | 0.0516 ± 0.0190 |
> | VIDD | **0.5885 ± 0.0368** | **0.1241 ± 0.0352** |
>
> >Q12. For protein experiments, I think there would be potential data mismatch between EvoDiff and VIDD. Would you please clarify it?
>
> Thanks for your question. We believe there is no dataset mismatch in the protein experiments. Our framework does not introduce any new training data: it relies solely on (i) a pretrained generative model (EvoDiff) and (ii) a reward model (as described in Algorithm 1). VIDD simply fine-tunes EvoDiff using reward feedback to obtain a reward-guided generator, and it does not access or require any additional sequence dataset.
>
> Thus, no external data are introduced, and therefore no dataset mismatch arises. If this does not fully address the concern, we would be happy to provide further clarification on the reviewer’s specific interpretation of “data mismatch.”

---

> ### Author Response · Authors · 2025-11-23
>
> >W2. Only limited baseline works were discussed and compared. Some baselines from discrete flow matching/diffusion on biological sequence design are of high correlation with this work [1-4].
>
> We appreciate the reviewer’s suggestion to broaden the baseline discussion and include additional discrete flow/diffusion works in biological sequence modeling. Below we clarify how these directions relate to VIDD.
>
> [1] Peptune suggests a new inference-time search method to search for an optimized therapeutic peptide, which is in the category of inference-time algorithm, and is orthogonal with our fine-tuning method VIDD. We place this work in Section 1.1 to show it’s another inference-time work.
>
> [2] CTRL performs RL fine-tuning under a purely offline setting and therefore requires a curated conditional dataset $\{(x,c)\}$. In contrast, VIDD optimizes directly against a reward model and does not require any conditioning data, enabling de novo design without offline datasets. Therefore, CTRL cannot operate in our settings where rewards are available but no labeled data exist.
>
> [3] GLID$^{2}$E is correlated with our work, so we additionally report its performance in Table below (and Table 8 in paper) as a baseline for the DNA sequence generation task. The method serves as a strong comparison point based on the results reported in its original paper. Nevertheless, VIDD achieves the highest gain in Pred-Activity, highlighting its strong reward-oriented fine-tuning capability. Regarding 3-mer correlation, although VIDD departs from the pretrained model’s k-mer statistics, this divergence does not represent failure. Instead, it suggests that VIDD discovers novel yet functionally meaningful sequences beyond the conventional motif patterns learned by the pretrained model.
>
>
> | Method              | Pred-Activity ↑          | ATAC-Acc ↑                     | 3-mer Corr ↑ | Log-Lik ↑          |
> |---------------------|--------------------------|--------------------------------|--------------|--------------------|
> | DRAKES w/o KL       | 6.44 ± 0.04              | 0.825 ± 0.028                | 0.307        | -281 ± 0.6         |
> | DRAKES          | 5.61 ± 0.07              | **0.925 ± 0.006**              | **0.887**    | -264 ± 0.6         |
> | GLID²E              | 7.35 ± 0.07              | 0.906 ± 0.003                  | 0.490        | -240 ± 14.2        |
> | VIDD            | **8.28 ± 0.18**          | 0.820 ± 0.384                  | 0.162        |**-198 ± 8.6**       |
>
> [4] STG-Flow addresses a fundamentally different problem. It is designed for conditional generation, where training data provide explicit mappings between sequences and conditions, and the method relies on differentiable conditional signals for learning. In contrast: (1) VIDD does not require any differentiable condition; (2) VIDD does not create or depend on any dataset for conditional supervision; and (3) VIDD is not learning a mapping but instead directly maximizes a reward objective. As a result, STG-Flow cannot operate under the VIDD problem setting, which assumes no accessible conditional data and non-differentiable reward functions.
>
>
>
>
>
>
>
>
> [1] Li X, Zhao Y, Wang C, et al. Derivative-free guidance in continuous and discrete diffusion models with soft value-based decoding[J]. arXiv preprint arXiv:2408.08252, 2024.
>
> [2] Rector-Brooks J, Hasan M, Peng Z, et al. Steering masked discrete diffusion models via discrete denoising posterior prediction[J]. arXiv preprint arXiv:2410.08134, 2024.

---

> > ### Comment · Reviewer_dS5g · 2025-11-26
> >
> > The authors addressed my concerns. I will raise my score to 6.
> >
> > Also, I appreciate the additional experimental results proposal. They would be a good source for this task and field.
> >
> > Best,
> >
> > Reviewer

---

> > > ### Author Response · Authors · 2025-11-26
> > >
> > > Thank you very much for the updated score and for your encouraging feedback. We appreciate your time and thoughtful review throughout the process!

---

### Official Review · Reviewer_57LE · 2025-10-31

**Soundness:** 3
**Presentation:** 4
**Contribution:** 3
**Rating:** 6
**Confidence:** 4

**Summary:**

This paper presents **VIDD (Value-guided Iterative Distillation for Diffusion models)** — a reward-guided fine-tuning framework for diffusion models in biomolecular design tasks where reward functions are often non-differentiable.
VIDD reformulates RL-style fine-tuning as **off-policy, value-weighted maximum likelihood estimation** with a **forward KL** objective, addressing instability and mode collapse common in PPO-like methods.
It iteratively distills soft-optimal denoising policies through three phases: off-policy roll-in, reward-weighted roll-out, and KL-based distillation.
Experiments across **protein**, **DNA**, and **molecular** design tasks show strong and stable improvements over DDPO, DDPP, and other baselines, including differentiable settings.

**Strengths:**

1. **Principled Distillation Framework:** The paper introduces a principled distillation-based alternative to PPO-style fine-tuning, effectively addressing instability issues commonly seen in on-policy reinforcement learning for diffusion models.

2. **Handling Non-differentiable Rewards:** The proposed method applies to realistic biomolecular tasks where reward gradients are unavailable, bridging diffusion modeling with practical scientific design scenarios.

3. **Stability via Lazy Updates:** The lazy-update strategy allows training to use mostly stable, older data while periodically refreshing the roll-out policy to capture model improvements — striking a balance between stability and adaptivity.

4. **Comprehensive Evaluation:** The experiments are comprehensive, covering multiple scientific domains (protein, DNA, molecule) and providing ablations and comparisons with strong baselines.

**Weaknesses:**

1. **Heuristic Value Approximation:** The soft-value approximation via a single forward reward prediction (Algorithm 1, line 8) may introduce bias or variance, but the paper lacks quantitative analysis of its impact.
2.  **Limited Theoretical Rigor:** The forward-KL interpretation is intuitive yet only algebraically shown (Appendix B) without a formal derivation or stability proof.
3. **Comparisons on Differentiable Rewards:** For differentiable cases (e.g., DNA enhancer tasks), comparisons with gradient-based fine-tuning methods such as DRAKES are limited; more systematic evaluation would strengthen claims of generality.
4. **Overlap with Prior Work:** The framework is conceptually related to value-weighted MLE and offline RL distillation and the paper could better position its novelty beyond these precedents.

**Questions:**

1. How sensitive is VIDD to the choice of α (temperature) and lazy update interval K?

2. How does the algorithm behave when the reward oracle is noisy or misaligned (e.g., false-positive docking scores)? Does it still remain stable, or collapse due to biased reward?

3. Can VIDD be combined with differentiable reward gradients (hybrid)? For differentiable domains, could both gradient and value-weighted distillation terms coexist?

---

> ### Author Response · Authors · 2025-11-23
>
> Thanks for the valuable feedback and insightful comments. We address each concern in detail below, and the corresponding revisions have been updated in the manuscript. All changes are highlighted in blue italics for clarity.
>
> > W1. Heuristic Value Approximation: The soft-value approximation via a single forward reward prediction (Algorithm 1, line 8) may introduce bias or variance, but the paper lacks quantitative analysis of its impact.
>
> Thanks for pointing this out. Posterior mean value estimation can in principle introduce bias or variance, since it approximates $\mathbb{E}[x_0 \mid x_t]$ with a single decoded sample. Our goal is not to eliminate this error, but to examine whether such approximation is sufficient to successfully guide reward-driven fine-tuning. Our estimator takes advantage of a core characteristic of diffusion models: they are trained to map noisy inputs back toward a clean sample. Consequently, the model’s prediction $ \hat{x}_0(x_t) $ serves as a practical proxy for the conditional expectation $ \mathbb{E}[x_0 \mid x_t] $, leading to the approximation $ v_t(x_t) \approx r(\hat{x}_0(x_t)) $. Similarly, this kind of idea also appeared in previous works (SVDD-PM [1]).
>
> To provide quantitative evidence, we now include a comparison between posterior-mean decoding and temperature-based Monte Carlo estimation (see Tables below, more details are in Table 21 and Table 22 in Appendix F). We can observe that posterior mean estimation yields higher reward performance compared with Monte Carlo estimator, even Monte Carlo estimator will have N times computation on reward calculation.
>
> Thus, VIDD’s value approximation may involve some estimation errors, but it is still effective enough as a practical estimator for reward-guided fine-tuning, and we have quantified its impact in the revised version.
>
> PD-L1
> |                       | Reward↑              | ipTM↑               | pLDDT↑              | Radius↓                 | Diversity↑ |
> |-----------------------|----------------------|---------------------|---------------------|--------------------------|-------------|
> | Posterior mean    | **0.9079 ± 0.0237**  | **0.8182 ± 0.0213** | **0.8720 ± 0.0421** | -0.1232 ± 0.1066     | **0.5539**  |
> | Monte carlo estimation (M=4) | 0.7758 ± 0.0620  | 0.7105 ± 0.0603     | 0.4993 ± 0.0359     | **-0.7711 ± 0.1049**         | 0.5155      |
>
>
> IFNAR2
> |                       | Reward↑              | ipTM↑               | pLDDT↑              | Radius↓                 | Diversity↑ |
> |-----------------------|----------------------|---------------------|---------------------|--------------------------|-------------|
> | Posterior mean    | **0.5120 ± 0.1093**  | **0.5090 ± 0.1079** | 0.4711 ± 0.0490 | 2.2039 ± 1.9989      | 0.5176  |
> | Monte carlo estimation (M=4) | 0.4149 ± 0.0936  | 0.3474 ± 0.0906     | **0.6892 ± 0.0617**     | **0.0703 ± 0.2881**          | **0.5195**      |
>
> > W2. Limited Theoretical Rigor: The forward-KL interpretation is intuitive yet only algebraically shown (Appendix B) without a formal derivation or stability proof.
>
> Thanks for pointing it out. We have revised Appendix B to explicitly show the missing intermediate derivation steps behind the forward-KL formulation, which makes the proof easier to follow without changing its theoretical content.
>
> > W3. Comparisons on Differentiable Rewards: For differentiable cases (e.g., DNA enhancer tasks), comparisons with gradient-based fine-tuning methods such as DRAKES are limited; more systematic evaluation would strengthen claims of generality.
>
> Thanks for raising this point. We agree that when the reward function is differentiable, gradient-based methods such as DRAKES can directly optimize $max_{x\sim P^{\theta}}[r]$ through backpropagation. However, this assumption rarely holds in realistic biomolecular settings: many reward models (e.g., AlphaFold, ESMFold, etc) produce scores via large models (or even experiments) whose gradients are either inaccessible or expensive to compute. As model size becomes larger, computing derivatives is harder. In such cases, the reward can only be used as a sample-level signal, acting as an importance weight rather than a differentiable supervision.
>
> Therefore, our focus is on providing a fine-tuning framework that does not rely on reward differentiability, enabling optimization when only reward queries are available. We include results of DRAKES to illustrate how VIDD compares even when reward gradients are available, not make it as a practical baseline to directly compare. However, focusing on a systematic evaluation under differentiable-only rewards would constrain the scope to a narrow subset of design problems and would not reflect the broader biomolecular settings VIDD targets, where reward functions are typically non-differentiable or black-box.

---

> ### Author Response · Authors · 2025-11-23
>
> > W4. Overlap with Prior Work: The framework is conceptually related to value-weighted MLE and offline RL distillation and the paper could better position its novelty beyond these precedents.
>
> Thanks for the comment. While value-weighted MLE and offline RL distillation weight trajectories by reward, VIDD differs in where and how the weighting is applied. Instead of placing reward only on the final sample $x_0$ and optimize the whole trajectory from initial state to final state​, VIDD optimizes intermediate diffusion states at time $t$ using an approximate soft value $v_t$​ (Eq. (3)), providing step-wise training signals throughout the denoising process. This reward-compatiable and diffusion-aware formulation is not present in prior value-weighted or offline distillation methods.
>
> > Q1. How sensitive is VIDD to the choice of α (temperature) and lazy update interval K?
>
> We add additional results for varying both $\alpha$ and the lazy update interval $K$ in Tables 9, 10, 16, 17, 18, 19, 20. Overall, both parameters affect performances. For $\alpha$, we consistently observe that $\alpha = 1.0$ yields the best results, as it preserves the original reward distribution without distortion. For $K$, tuning is beneficial: $K=1$ tends to underperform due to overly frequent updates, and we recommend starting around $K \approx 5$ and increasing gradually if needed.
>
> > Q2. How does the algorithm behave when the reward oracle is noisy or misaligned (e.g., false-positive docking scores)? Does it still remain stable, or collapse due to biased reward?
>
> To assess robustness, we manually inject 10% noise into the reward values and report the results in Table 23 and Table 24 (Appendix G). We observe a clear degradation in performance when noise is introduced. This outcome is reasonable because VIDD does not yet include any robustness-enhancing mechanisms. We discuss this limitation in Appendix C.2 and suggest that accurate reward signals are the key for successfully fine-tuning the generative model. We also identify robustness-aware reinforcement learning as a promising direction to improve VIDD under noisy or biased reward settings.
>
>
> PD-L1
> |               | Reward↑              | ipTM↑               | pLDDT↑              | Radius↓                | Diversity↑ |
> |---------------|----------------------|---------------------|---------------------|-------------------------|-------------|
> | noise=0.0 | **0.9079 ± 0.0237**  | **0.8182 ± 0.0213** | **0.8720 ± 0.0421** | **-0.1232 ± 0.1066**    | **0.5539**  |
> | noise=0.1     | 0.4744 ± 0.1172      | 0.4570 ± 0.1192     | 0.4306 ± 0.0255     | 1.2802 ± 0.9287         | 0.5154      |
>
> IFNAR2
> |               | Reward↑              | ipTM↑               | pLDDT↑              | Radius↓                | Diversity↑ |
> |---------------|----------------------|---------------------|---------------------|-------------------------|-------------|
> | noise=0.0 | **0.5120 ± 0.1093**  | **0.5090 ± 0.1079** | **0.4711 ± 0.0490** | 2.2039 ± 1.9989     | 0.5176  |
> | noise=0.1     | 0.1970 ± 0.0217      | 0.1446 ± 0.0160     | 0.4692 ± 0.0620     | **-0.2763 ± 0.2740**        | **0.5234**  |
>
>
> > Q3. Can VIDD be combined with differentiable reward gradients (hybrid)? For differentiable domains, could both gradient and value-weighted distillation terms coexist?
>
> Current VIDD design is not directly compatible with gradient-based reward optimization. When the reward is differentiable, the objective $\max_{x \sim p_{\theta}}[r]$ allows gradients to flow from the reward model into the generator parameters, so the generator can be optimized directly. In contrast, VIDD assigns weights based on the soft value at intermediate states $x_t$​, reweighting samples rather than backpropagating through the reward model. As a result, reward gradients cannot be directly combined with VIDD’s value-based weighting.
> In summary, when reward gradients are available and cheap to compute, gradient-based optimization is likely preferable; when reward functions are non-differentiable or expensive, gradient-free value distillation like VIDD is a more practical choice.
>
>
>
> [1] Li X, Zhao Y, Wang C, et al. Derivative-free guidance in continuous and discrete diffusion models with soft value-based decoding[J]. arXiv preprint arXiv:2408.08252, 2024.

---

### Author Response · Authors · 2025-12-03
**Discussion Summary**

We sincerely thank all reviewers and ACs for their time and constructive feedback under this year’s unique circumstances. Given the terminated discussion period, we provide a concise summary to support your final evaluation.

### **1. Score Updates Following the Rebuttal**

Before the OpenReview leakage on Nov 27, two reviewers who actively engaged with our rebuttal explicitly raised their scores: Reviewer dS5g: **4 → 6**, Reviewer AJss: **6 → 8**.

Both reviewers stated that their concerns had been fully resolved and raised
their scores accordingly. These updates were made on Nov 25, quickly following our
full rebuttal on Nov 23.

### **2. Details of the Concerns and Our Additional Clarifications**

Across all four reviews, the primary concerns consistently fall into future-direction categories, including:
- Variants of soft-value approximation
- Hyperparameter sensitivity ($K$, $\alpha$, $\beta_s$)
- Robustness to reward noise
- Theoretical clarity and positioning

We have added detailed discussion and new experiments addressing all of the
above points. These concerns reflect extensions and variants rather than
challenges to the core method or main results: fine-tuning diffusion models
via our novel iterative distillation of target policies remains sound and effective.

To support this, our rebuttal introduced substantial new analyses:

- New experiments on value-approximation variants to analyze approximation behavior
- 7 new detailed ablation tables on hyperparameter analysis for $K$, $\alpha$, $\beta_s$
- New experiments on reward-noise perturbation
- Additional clarification on theoretical proof

The two reviewers who raised their scores confirmed that these additions
resolved their concerns. The remaining reviewers’ comments are all aligned
with the same themes and are also fully covered by these analyses, although
they were unable to update their evaluations due to the discussion freeze.



### **3. Final Remarks**

We deeply appreciate the efforts of all reviewers and both ACs throughout this
process. We also understand that the newly assigned AC is reviewing the paper from
scratch, and we are glad to provide any further clarification if helpful.


Thank you again for your time and consideration.

Best regards,

The Authors

---

### Meta-Review · Area_Chair_Gx1E · 2026-01-06

**Summary:**

The submitted manuscript raised several concerns, which can be broadly classified into two categories.
- Limited empirical study. For instance, the empirical study lacks important ablations, empirical analysis of the bias introduced by soft-value approximation, some baselines, robustness to noisy rewards, and a comparison in the case when the rewards are differentiable (Reviewers 57LE, dS5g, F9Zw, AJss).
- Missing derivations and theoretical analysis (Reviewer 57LE, dS5g).

**Reviewer Concerns:**

The authors provided an extensive response answering in detail all the concerns of the reviewers. The paper's rebuttal is remarkable on both sides: the reviewers offered constructive suggestions to strengthen the paper's empirical evaluation, and the authors implemented these suggestions, significantly improving the manuscript.

**Reviewer Scores:**

The rebuttal phase has all the evidence that the initial scores would be significantly increased.

---

### Decision · Program_Chairs · 2026-01-26

Accept (Poster)